# Rapid intra-host diversification and evolution of SARS-CoV-2 in advanced HIV infection

Sung Hee Ko [1,9], Pierce Radecki [1,9], Frida Belinky[1], Jinal N. Bhiman [2,3], Susan Meiring [2], Jackie Kleynhans [2,4], Daniel Amoako [2,5], Vanessa Guerra Canedo[1], Margaret Lucas [1], Dikeledi Kekana[2], Neil Martinson[6,7], Limakatso Lebina[6], Josie Everatt[2], Stefano Tempia [2,4], Tatsiana Bylund[1], Reda Rawi [1], Peter D. Kwong [1], Nicole Wolter [2,8], Anne von Gottberg [2,8], Cheryl Cohen [2,4,10] & Eli A. Boritz [1,10] ✉

Previous studies have linked the evolution of severe acute respiratory syndrome coronavirus-2 (SARS-CoV-2) genetic variants to persistent infections in people with immunocompromising conditions, but the processes responsible for these observations are incompletely understood. Here we use high-throughput, single-genome amplification and sequencing (HT-SGS) to sequence SARS-CoV-2 spike genes from people with HIV (PWH, $n = 22$) and people without HIV (PWOH, $n = 25$). In PWOH and PWH with CD4 T cell counts (i.e., CD4 counts) ≥ 200 cells/μL, we find that most SARS-CoV-2 genomes sampled in each person share one spike sequence. By contrast, in people with advanced HIV infection (i.e., CD4 counts < 200 cells/μL), HT-SGS reveals a median of 46 distinct linked groupings of spike mutations per person. Elevated intra-host spike diversity in people with advanced HIV infection is detected immediately after COVID-19 symptom onset, and early intra-host spike diversity predicts SARS-CoV-2 shedding duration among PWH. Analysis of longitudinal timepoints reveals rapid fluctuations in spike sequence populations, replacement of founder sequences by groups of new haplotypes, and positive selection at functionally important residues. These findings demonstrate remarkable intra-host genetic diversity of SARS-CoV-2 in advanced HIV infection and suggest that adaptive intra-host SARS-CoV-2 evolution in this setting may contribute to the emergence of new variants of concern.

While mounting evidence suggests that SARS-CoV-2 genetic variants emerge preferentially in immunocompromised individuals[1–4], the processes of intra-host evolution that produce these variants are incompletely understood. Multiple studies have documented new SARS-CoV-2 mutations in people with HIV (PWH), with conditions requiring immunosuppressive therapy, and/or with B cell deficiencies[5–12]. New mutations have typically been detected in these cases weeks or months after COVID-19 symptom onset, with one recent study[13] reporting that overall SARS-CoV-2 mutation rates were similar between short-term and persistent infections. These findings suggest a temporal threshold after which SARS-CoV-2 has accumulated enough mutations to evolve within the individual. However, many previous studies retrospectively characterizing persistent infections have been limited to case reports or small case series, and have not

closely examined the early stage of infection[3–11,14,15]. Equally important, while standard technologies can track SARS-CoV-2 consensus sequence changes and identify some minor variant mutations in genomic surveillance[16–19], advanced approaches that define intra-host virus genetic diversity and evolution at the single-genome level have not been widely used. Addressing these gaps may help elucidate how SARS-CoV-2 establishes persistent infection and generates new sequence variants.

To define the genetic diversity and evolutionary signatures among SARS-CoV-2 genomes in each individual, we have developed a high-throughput, single-genome amplification and sequencing (HT-SGS) approach that combines unique barcoding of virus genomes with long-read sequencing to produce up to ~$10^3$ single-copy sequences per sample[20]. Here we used HT-SGS of the full-length spike gene to analyze a unique cohort of clinically diverse PWH and people without HIV (PWOH) sampled from onset to clearance of SARS-CoV-2 infection[21,22]. Through longitudinal analysis of SARS-CoV-2 spike sequences together with detection of anti-spike antibody binding, we find unique aspects of SARS-CoV-2 evolution in people with advanced, poorly controlled HIV infection that markedly increase the risk for generation of new SARS-CoV-2 variants in these individuals.

## Results

### Longitudinal sampling of PWH and PWOH

We investigated intra-host evolution of SARS-CoV-2 during persistent infection using longitudinal sample sets from 22 PWH and 25 PWOH. These individuals had participated in cohort studies of people with COVID-19 diagnosed either as hospital inpatients or as outpatients between May 1, 2020 and December 31, 2020 (hospitalized cohort)[21] or between October 2, 2020 and September 30, 2021 (outpatient cohort)[22]. From the hospitalized cohort, we included a subgroup of 10 PWH with peripheral blood CD4 T cell counts (i.e., CD4 counts) < 200 cells/µL who had high initial SARS-CoV-2 RNA levels in respiratory samples (rRT-PCR cycle threshold [Ct] < 30) (Supplementary Fig. 1; Supplementary Data 1). Remaining participants from the hospitalized cohort included 5 PWH for whom CD4 counts were not available, 3 PWH with CD4 counts ≥ 200 cells/µL, and 7 PWOH. From the outpatient cohort we included 2 PWH with CD4 counts < 200 cells/µL, 2 PWH with CD4 counts ≥ 200 cells/µL, and 18 PWOH (Supplementary Fig. 1a; Supplementary Data 1). Notably, among the 12 PWH with CD4 counts < 200 cells/µL, 6 had plasma HIV RNA > $10^5$ copies/mL, and 4 had no plasma HIV RNA level documented (Supplementary Data 1). Upper respiratory tract samples were available from these individuals beginning at study enrollment, at a median of 4 days (IQR 3-8) after the onset of COVID-19 symptoms, and every second day (hospitalized cohort) or three times weekly (outpatient cohort) thereafter until the cessation of SARS-CoV-2 RNA shedding. As described previously in the hospitalized cohort[21], PWH with CD4 counts < 200 cells/µL who had high initial SARS-CoV-2 RNA levels in respiratory specimens often experienced prolonged SARS-CoV-2 RNA shedding (Supplementary Fig. 1b).

### HT-SGS vs. standard whole-genome sequencing

SARS-CoV-2 spike sequences in upper respiratory tract swab samples from these individuals were determined by HT-SGS of the full-length spike gene (Supplementary Data 2). This approach detected mutations that were present in as few as 0.45% of amplifiable virus genomes per sample (Supplementary Fig. 2a). Control experiments using mixtures of synthetic RNAs supported the validity of minor variant spike sequences detected at this level in HT-SGS data (Supplementary Fig. 3). By considering the unique combinations of called mutations (i.e., haplotypes) that are supported by multiple single-genome sequences (SGS), HT-SGS demonstrates mutational linkage patterns across the 3.8-kilobase spike region that are not detectable by short-read whole-genome sequencing (WGS) (Supplementary Figs. 2b–d). Additional

controls using mixtures of synthetic RNAs verified that haplotypes detected by HT-SGS accurately represented sequences present in sample material, without significant evidence of technical recombination artifacts (Supplementary Fig. 4). By defining and quantifying spike gene haplotypes in each sample at the level of single genomes, HT-SGS provides minimum estimates of intra-host population diversity and enables downstream analysis of evolutionary relationships among viruses in each person (Supplementary Fig. 2b-d).

### Intra-host SARS-CoV-2 genetic diversity in PWH and PWOH

We used HT-SGS to analyze 184 samples from the 47 study participants, resulting in 70,968 SGS from PWH and 29,824 SGS from PWOH. These SGS included 431 different single-nucleotide variations (SNVs) or deletions that together defined 831 spike gene haplotypes. As shown in Fig. 1a, strikingly high numbers of spike haplotypes were detected in some PWH. This was especially true for PWH with CD4 counts < 200 cells/µL (sky blue tree sections, Fig. 1a), in whom a median of 46 haplotypes/person (IQR 14-114/person) were detected over the course of infection. Analysis of very rare SNVs and deletions indicated the presence of additional haplotypes at levels below reportable limits at the given sampling depth, particularly in PWH with CD4 counts < 200 cells/µL (Supplementary Fig. 5a). Considering each sample timepoint separately to avoid inflating genetic-distance-based diversity calculations in prolonged infections, we found higher intra-host diversity in PWH with CD4 counts < 200 cells/µL than in PWH with higher CD4 counts or in PWOH as measured by normalized Shannon entropy, average pairwise genetic distance, and total numbers of haplotypes detected (Fig. 1b). These diversity measures were similar between PWH with higher CD4 counts and PWOH. While the number of haplotypes identified in each sample was positively correlated with the number of SGS obtained from that sample (Supplementary Fig. 5b), ratios of haplotypes identified per SGS obtained were nonetheless significantly higher in PWH with CD4 counts < 200 cells/µL than in the other subgroups (Supplementary Fig. 5c). Moreover, the differences in SARS-CoV-2 spike gene diversity between subgroups described above were also observed in an analysis limited to the hospitalized cohort (Supplementary Fig. 5d). We conclude that HT-SGS revealed a markedly elevated intra-host diversity of SARS-CoV-2 spike haplotypes among people with advanced HIV infection.

To relate SARS-CoV-2 sequence variation detected in our study participants to the variation among viruses circulating contemporaneously in the same geographic region, we compared spike genetic divergence from Wuhan-Hu-1 (Hu-1, GenBank Accession NC_045512.2, nucleotide coordinates 21563-25384) between SGS from the hospitalized cohort and matched public WGS data[23] (Fig. 1c). In data on the NCBI Virus database from infections in South Africa between April and October 2020, 91.7% of spike sequences showed 1-3 mutations relative to Hu-1. In contrast, 31% of spike SGS from PWH with CD4 counts < 200 cells/µL had 4 or more mutations relative to Hu-1 ($p = 0.005$ for comparison of distributions in PWH with CD4 counts < 200 cells/µL vs. NCBI, Friedman test with Dunn's multiple comparisons test). Furthermore, SARS-CoV-2 sequence diversification in PWH from both hospitalized and outpatient cohorts was associated with intra-host emergence of variants that mapped by Nextclade[24] to secondary Pango lineages (Fig. 1d). Among PWH with CD4 counts < 200 cells/µL, this analysis indicated a median of 3.5 lineages (range, 1-9) in each person. Secondary intra-host lineages thus identified in PWH with CD4 counts < 200 cells/µL included B.1.1.525, B.1.214.3, and C.2.1 at timepoints preceding the first reports of these lineages in global surveillance (1/5/2021, 12/14/2020, and 12/18/2020). No secondary Pango lineages were identified among intra-host sequences from PWOH, regardless of infecting variant or clinical cohort (Fig. 1d). We conclude that SARS-CoV-2 sequence divergence from ancestral detected by HT-SGS in people with advanced HIV infection significantly exceeded the divergence of geographically- and temporally matched circulating

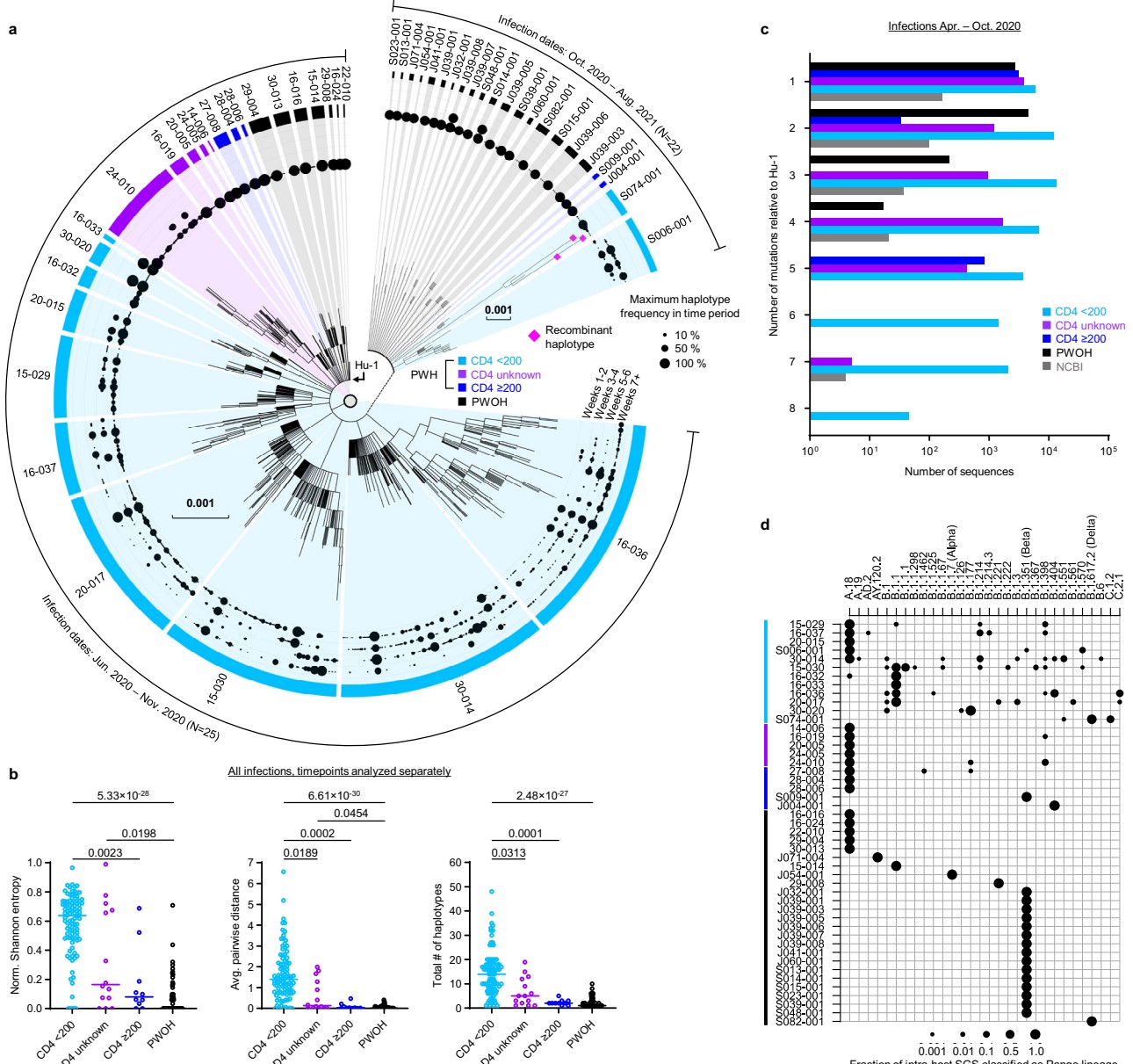

**Fig. 1 | Intra-host diversity of SARS-CoV-2 spike sequences in PWH and PWOH.** **a** Maximum-likelihood phylogenetic analysis of spike gene haplotypes detected in each participant. Trees were generated separately for each participant and then joined for visualization with Wuhan-Hu-1 (Hu-1). The maximum frequency of each haplotype during each two-week period is shown via the dot plot. Trees from hospitalized and outpatient cohorts are separated to reflect differences in infecting Pango lineages. Color coding of PWOH and PWH subgroups applies to all figures. **b** Comparison of spike genetic diversity among PWOH and subgroups of PWH. Individual samples from longitudinal sample sets in each person are represented by separate datapoints (sample size: $n = 96$ in CD4 < 200, $n = 10$ in CD4 ≥ 200, $n = 14$ in CD4 unknown, $n = 59$ in PWOH). Statistical significance was assessed by one-way ANOVA with multiple comparisons (Kruskal-Wallis test and Dunn's multiple comparisons test); $p$ values < 0.05 are shown. **c** Genetic divergence (number of

mutations, $y$-axis) from ancestral Wuhan-Hu-1 in SARS-CoV-2 spike sequences from PWH subgroups, PWOH, and matched public data. Numbers of sequences ($x$-axis) indicate total numbers of single-genome sequences (SGS) for all participants combined in PWOH and PWH subgroups, or total numbers of single-person consensus sequences obtained from the NCBI Virus database. Statistical significance was assessed by one-way ANOVA with multiple comparisons (Friedman test and Dunn's multiple comparisons test); statistically significant $p$ values were detected for the following comparisons: CD4 < 200 vs. NCBI ($p = 0.005$), CD4 < 200 vs. CD4 ≥ 200 ($p = 0.002$), and CD4 < 200 vs. PWOH ($p = 0.006$). **d** Analysis of secondary intra-host Pango lineages. Distinct lineages are indicated as columns; individual participants are indicated as rows. Dots represent relative frequencies of individual lineages among all SGS from each participant. Source data are provided as a Source Data file.

sequences, in some cases anticipating the later emergence of SARS-CoV-2 genetic variants in the general population.

## Spike evolution over time

To define the kinetics of intra-host SARS-CoV-2 evolution in our study cohort, we analyzed longitudinal patterns in spike HT-SGS data from PWH and PWOH. We observed that normalized Shannon entropy,

average pairwise genetic distance, and total haplotype numbers at the first sample timepoint after symptom onset were significantly higher in PWH with CD4 counts < 200 cells/µL than in PWOH (Fig. 2a) even though days between symptom onset and the first sample timepoint were not significantly different between groups (PWH with CD4 counts < 200 cells/µL median 4.5 days, IQR 4-13.5; PWOH median 3 days, IQR 2-5; $p = 0.1526$, Kruskal-Wallis). Higher initial spike gene diversity

predicted longer SARS-CoV-2 RNA shedding duration among PWH (Fig. 2b) but not among PWOH (Supplementary Fig. 5e). All measures of intra-host spike gene diversity increased significantly over time among PWH with CD4 counts < 200 cells/µL, reflecting longer infections in participants with high initial diversity as well as rising diversity in some participants (Fig. 2c). In a longitudinal analysis of the most abundant spike haplotypes in each person, rapid fluctuations in

detected frequencies observed in most PWH with CD4 counts < 200 cells/µL (Supplementary Fig. 6a) contrasted sharply with the persistence of a single predominant haplotype throughout the course of infection in every PWOH (Supplementary Fig. 6b). Haplotype abundance fluctuations were often associated with progressively lower frequencies of the presumptive founder haplotype over time in each PWH with CD4 count < 200 cells/µL (Fig. 2d). The use of Jensen-

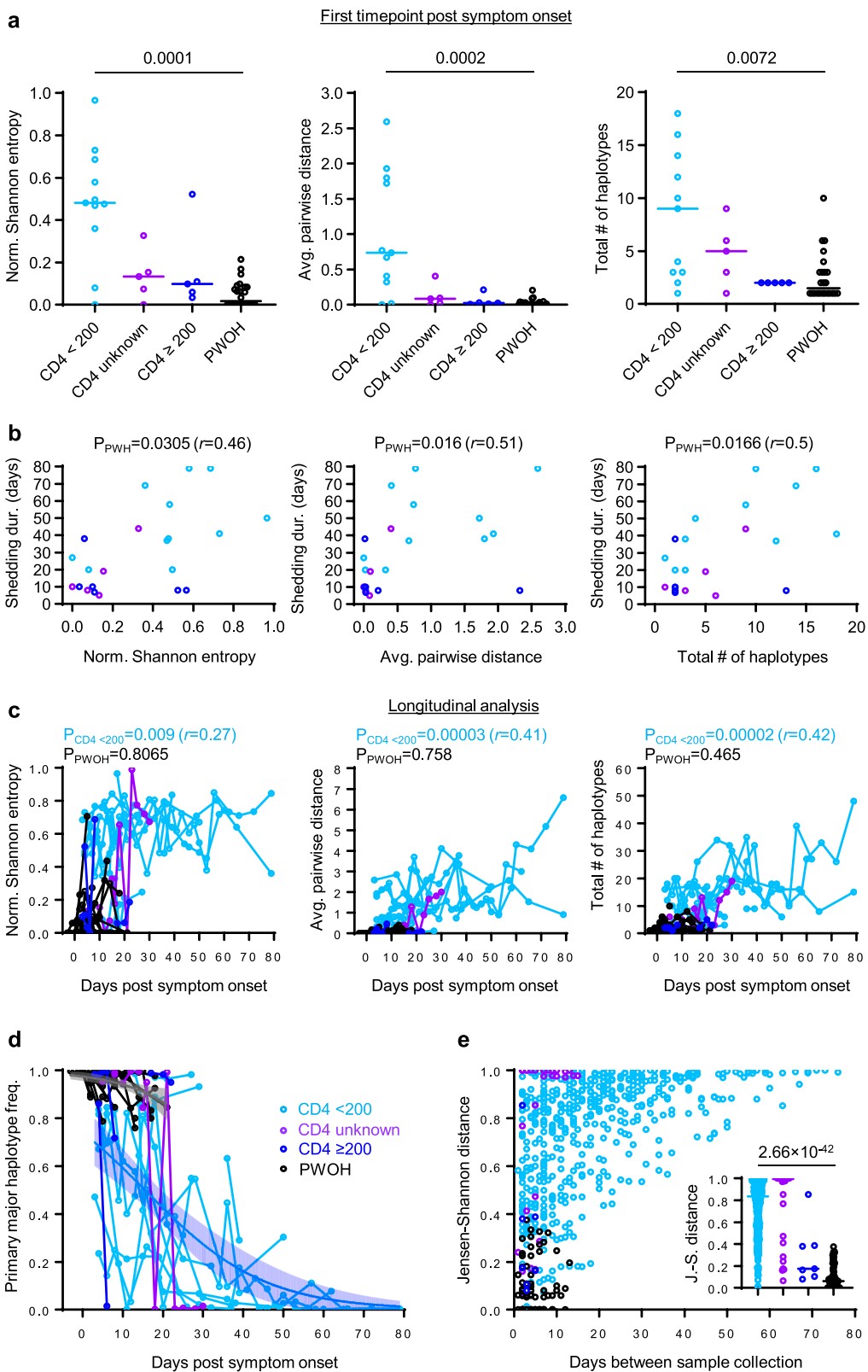

**Fig. 2 | Longitudinal analysis of intra-host spike evolution in PWH and PWOH.** **a** Comparison of spike genetic diversity among PWOH and subgroups of PWH at the first sample timepoint. Each datapoint represents an individual sample of each person (sample size: $n = 11$ in CD4 < 200, $n = 5$ in CD4 ≥ 200, $n = 5$ in CD4 unknown, $n = 24$ in PWOH). Statistical significance was assessed by one-way ANOVA with multiple comparisons (Kruskal-Wallis test and Dunn's multiple comparisons test); $p$ values < 0.05 are shown. **b** Correlations between measurements of spike diversity at the first sample timepoint and SARS-CoV-2 RNA shedding duration in all PWH analyzed together. Each datapoint represents an individual sample of each person. Statistical significance was assessed by two-tailed Spearman's rank correlation test; $p$ values are shown. **c** Longitudinal changes in measurements of spike diversity in PWH subgroups and PWOH. Individual samples from longitudinal sample sets in each person are represented by separate datapoints. Two-tailed Spearman $p$ values

for correlations between measurements of spike diversity and time of sampling (days post symptom onset) are shown for PWH with CD4 counts < 200 cells/μL and PWOH. **d** Longitudinal changes in the intra-host frequency in PWH subgroups and PWOH of the primary major haplotype (i.e., the most abundant haplotype detected in the individual's first sample timepoint). Blue and gray curves indicate logistic regressions of mean frequency declines for PWH with CD4 counts < 200 cells/μL and PWOH; shaded areas (error bands) indicate bootstrapped 95% confidence intervals of the mean. **e** Pairwise similarity analysis (Jensen-Shannon distance) of virus populations for all pairs of sample timepoints in each participant. The inset panel compares Jensen-Shannon distances by participant subgroup for sample pairs collected ≤ 14 days apart. Statistical significance was assessed by one-way ANOVA with multiple comparisons (Kruskal-Wallis test and Dunn's multiple comparisons test). Source data are provided as a Source Data file.

Shannon distance to quantify global changes in detectable population haplotype composition over time in all pairs of sample timepoints in each person revealed large changes in PWH with CD4 counts < 200 cells/μL (Fig. 2e). Although the magnitude of these changes was correlated with the time between samples, large changes were observed in PWH with CD4 counts < 200 cells/μL even over short time intervals (main panel and inset panel, Fig. 2e). Taken together, these findings show that the development of elevated SARS-CoV-2 spike gene diversity in PWH with CD4 counts < 200 cells/μL encompasses 1) high early diversity, beginning shortly after COVID-19 symptom onset, and 2) marked changes in the population of sequences detected in each person over time.

### Analysis of transmitted SARS-CoV-2 spike diversity

The diversity of spike sequences detected in some PWH at early timepoints after COVID-19 symptom onset raised the possibility of multiple founder sequences in these individuals. To address this possibility and to limit the chance that downstream analysis would erroneously treat phylogenetic relationships between separate founder clades as intra-host evolution, we performed single-linkage phylogenetic clustering to identify clades of haplotypes in each person that were separated by at least 5 mutations and were thus likely to have originated from distinct founders[25]. This analysis revealed evidence of multiple founder sequences in 2 PWH, both of whom had CD4 counts < 200 cells/μL (Supplementary Fig. 7a). No evidence for >1 founder sequence was detected in any PWOH. In the PWH with evidence for >1 founder sequence, recombination analysis for participant S074-001 revealed low frequencies of 3 distinct intra-host recombinant haplotypes between Delta (B.1.617.2) and C.1.2 variant lineages (Supplementary Figs. 7a, b). Recombinant haplotypes were not detected in the other participants, potentially due to limited sensitivity of recombination analysis in the setting of low intra-host genetic diversity. Nevertheless, we conclude that infections with multiple SARS-CoV-2 founder sequences can be detected in PWH, and this may contribute to the generation of further SARS-CoV-2 genetic diversity through intra-host recombination.

### Spike mutations in PWH and PWOH

We investigated the nature of SARS-CoV-2 genetic diversity in PWH and PWOH by compiling all spike gene positions at which SGS were polymorphic over the course of each participant's infection. Intra-host polymorphisms (i.e., mutations found in <100% of SGS in the person) included synonymous SNVs, nonsynonymous SNVs, and deletions (Fig. 3). Synonymous SNVs were scattered across the spike gene at different positions in different participants (Figs. 3a, b; Supplementary Fig. 8a), reflecting their expected evolutionary neutrality, and were found at higher levels in PWH with CD4 counts < 200 cells/μL than in the other subgroups (Fig. 3c). These findings were consistent with elevated cumulative numbers of replicative cycles in PWH with CD4 counts < 200 cells/μL, as suggested by high initial virus RNA levels and prolonged shedding in these individuals[21]. At the same time, HT-SGS revealed

extensive intra-host nonsynonymous spike gene variation in the cohort. Many nonsynonymous mutations were deletions or nonsynonymous SNVs at three recurrently deleted sites in the NH$_2$-terminal domain (NTD)[26], with additional nonsynonymous SNVs in the receptor-binding domain (RBD), furin cleavage site, and membrane fusion regions, and with deletions in the fusion peptide (Fig. 3a, b). Although several PWH with higher CD4 counts and PWOH also showed nonsynonymous intra-host mutations (Fig. 3a, b; Supplementary Fig. 8a), the number of nonsynonymous intra-host mutations per person was significantly higher in PWH with CD4 counts < 200 cells/μL than in the other subgroups (Fig. 3d), likely a reflection of higher levels of virus replication and longer durations of infection in this subgroup. The intra-host nonsynonymous mutations detected in PWH with CD4 counts < 200 cells/μL overlapped significantly with the defining mutations of variants of concern (VOCs) Omicron BA.1, Alpha, Beta, and Delta (44.2% [23 of 52] of codons with nonsynonymous mutations in VOCs also mutated in PWH with CD4 counts < 200 cells/μL, vs. 12.1% [155 of 1273] of all codons in spike mutated in PWH with CD4 counts <200 cells/μL; Fisher's exact test $p < 0.0001$) (Fig. 3e). Moreover, structure-based calculations of solvent-accessible surface area (SASA)[27,28] for each amino acid residue in spike showed that, while synonymous mutations were not biased to residues with high or low SASA, nonsynonymous mutations were more commonly detected in solvent-accessible residues (Supplementary Figs. 8b, c). The strength of this bias was similar in residues mutated in VOCs (Supplementary Fig. 8b). Thus, HT-SGS of the spike gene showed that well-described mutations associated with worldwide SARS-CoV-2 evolution occurred commonly as intra-host mutations in PWH with CD4 counts < 200 cells/μL, in the context of a high total burden of mutations in these individuals.

### Selection analysis

We next asked whether patterns of spike gene evolution in PWH with CD4 counts < 200 cells/μL indicated adaptation of the virus to the host or were instead consistent with chance expansion of variant haplotypes bearing random mutations (i.e., genetic drift). We used the FUBAR algorithm[29] in conjunction with the maximum-likelihood phylogeny for each participant to identify codons at which the calculated ratio of nonsynonymous to synonymous mutations (dN/dS) supported positive selective pressure. To minimize false-positive selection analysis results owing to recurrent low-frequency mutations, we considered mutations identified by dN/dS to be under positive selection only if their intra-host frequency increased by > 20% during the individual's infection. This analytical approach demonstrated positive selection at one or more spike gene positions in 7 of the 12 PWH with CD4 counts < 200 cells/μL (see colored symbols, Fig. 4 and Supplementary Fig. 7a). Intra-host mutations were identified by selection analysis in these individuals in the signal peptide, NTD, RBD, subdomain 1 and 2, and S2 regions of the spike gene. In sharp contrast, no sites under positive selection were detected in either PWOH (Supplementary Fig. 9) or PWH with CD4 counts ≥ 200 cells/μL (Supplementary Fig. 10). One (1) site under positive selection was detected in a

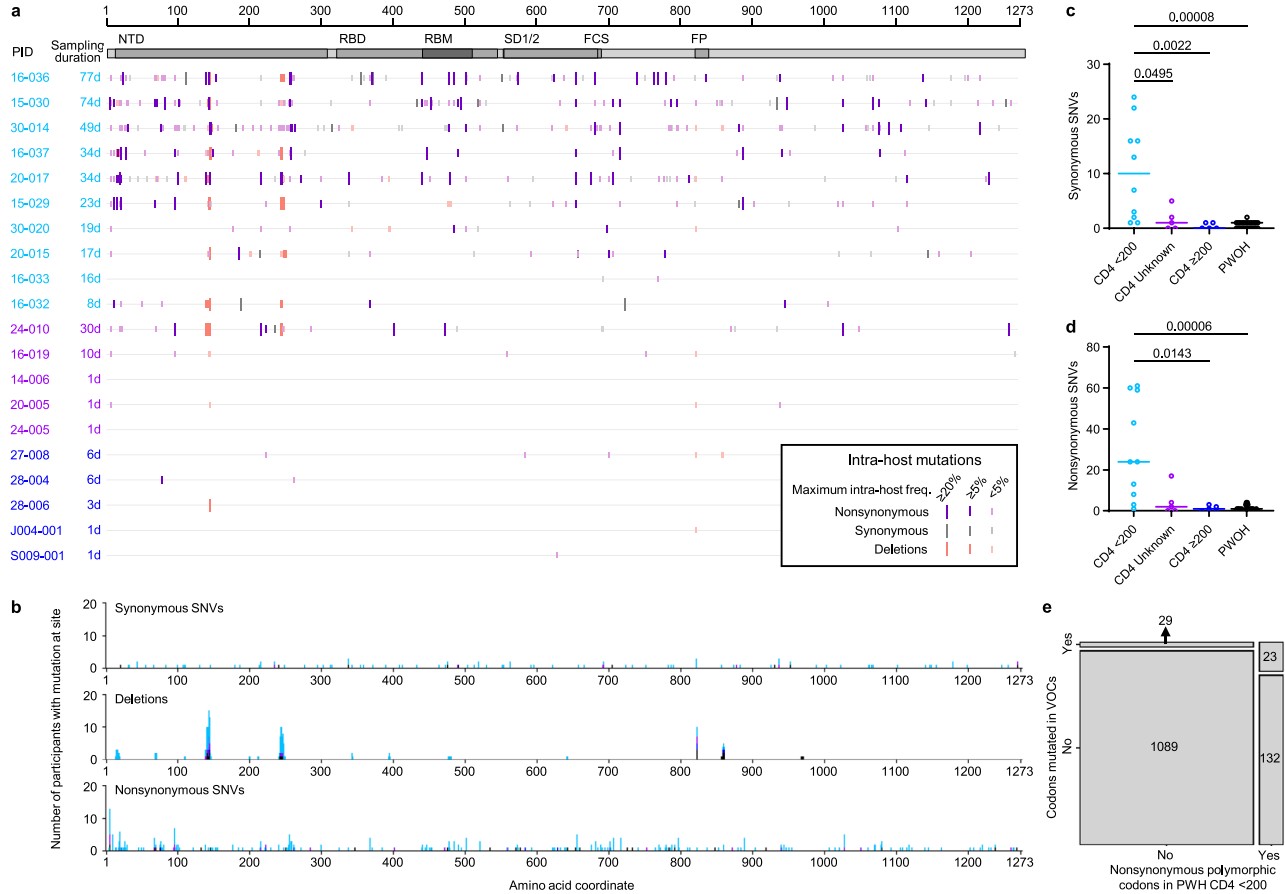

**Fig. 3 | Analysis of intra-host spike mutations in PWH and PWOH. a** Locations and types of intra-host spike mutations detected over all timepoints in PWH. The maximum measured frequency of each mutation is indicated by the size and color of the bar. Mutations observed in PWOH are shown in Supplementary Fig. 8a. **b** Total numbers of participants with synonymous (top), deletion (middle), and nonsynonymous (bottom) mutations detected at the indicated positions over all timepoints. Stacked bars are colored by participant subgroup. **c** and **d** Numbers of intra-host synonymous (**c**) and nonsynonymous (**d**) single-nucleotide variations (SNVs) by participant, compared among PWH subgroups and PWOH. Statistical significance was assessed by one-way ANOVA with multiple comparisons (Kruskal-Wallis test and Dunn's multiple comparisons test); $p$ values < 0.05 are shown (sample size: $n = 10$ in CD4 < 200, $n = 5$ in CD4 ≥ 200, $n = 5$ in CD4 unknown, $n = 25$ in PWOH). **e** Contingency analysis of codons in spike with nonsynonymous intra-host mutations in PWH with CD4 counts < 200 cells/μL vs. codons with non-synonymous mutations in the VOCs Alpha, Beta, Delta, and/or Omicron BA.1. The association between sites with nonsynonymous mutations in PWH with CD4 counts < 200 cells/μL and sites mutated in the VOCs was significant (two-sided Fisher's exact test, $p < 0.00001$). Source data are provided as a Source Data file.

PWH with unknown CD4 count who showed high spike diversity (participant 24-010, see Supplementary Fig. 10). Thus, selection analysis enabled by HT-SGS suggested that the intra-host population of SARS-CoV-2 spike sequences in PWH with CD4 counts < 200 cells/μL arose in part through adaptive evolution, and not purely through genetic drift.

**Selection and autologous antibody responses**

To investigate the intra-host selective forces that might drive SARS-CoV-2 genetic evolution in PWH with CD4 counts < 200 cells/μL, we cross-referenced spike gene phylogenetic and selection analyses with autologous antibody analysis for each participant. As described previously[21], serum binding to the ancestral SARS-CoV-2 spike remained undetectable in many of these individuals through the first 4 weeks of infection (see Fig. 4, purple traces, and Supplementary Fig. 11), consistent with other recent studies of SARS-CoV-2 humoral immunity in PWH[30,31]. Subsequently, in participants 16-036 and 15-030, serum binding to spike repeatedly exceeded the assay positivity threshold. These delayed but detectable responses were associated with positive selection for RBD mutations linked to immune escape, including presumptive convergent evolution of E484K (see Fig. 4, participant 16-036, red pentagons [#7]) and a virus genetic clade defined by L452R (see Fig. 4, participant 15-030, red pentagons [#4],

top clade on the tree). Increased frequency of L452R in participant 15-030 was associated with reduced SARS-CoV-2 RNA levels and a decrease in population diversity (i.e., entropy), followed by increased SARS-CoV-2 RNA levels and continued spike gene diversification (Supplementary Fig. 12). In other PWH with CD4 counts < 200 cells/μL, serum spike binding remained undetectable at all timepoints tested. These individuals did not show positive selection within the RBD but did show positive selection for mutations associated with changes in virus infectivity, including H655Y (see Fig. 4, participant 20-017, aqua circle [#5]) and P681H/R (see Fig. 4, participant 30-014, aqua circles [#1]). Notably, P681H was detected in 30-014 after an apparent population bottleneck (i.e., reduced SARS-CoV-2 RNA levels as reflected by Ct >35) and was followed by increased virus load and emergence of new spike variants (Supplementary Fig. 12). For many selected mutations in PWH with CD4 counts < 200 cells/μL, previous studies have provided evidence for antibody evasion and/or increased infectivity (Supplementary Table 1). Interestingly, selected mutations in the NTD that were associated with antibody evasion occurred in both the presence and the absence of detectable serum binding to spike (see Fig. 4, participants 20-017 and 15-029, blue hexagons), suggesting that alterations to this region could have important functional impacts beyond humoral immune evasion. Considering all pairs of successive timepoints in PWH with CD4 counts < 200 cells/μL, we found that

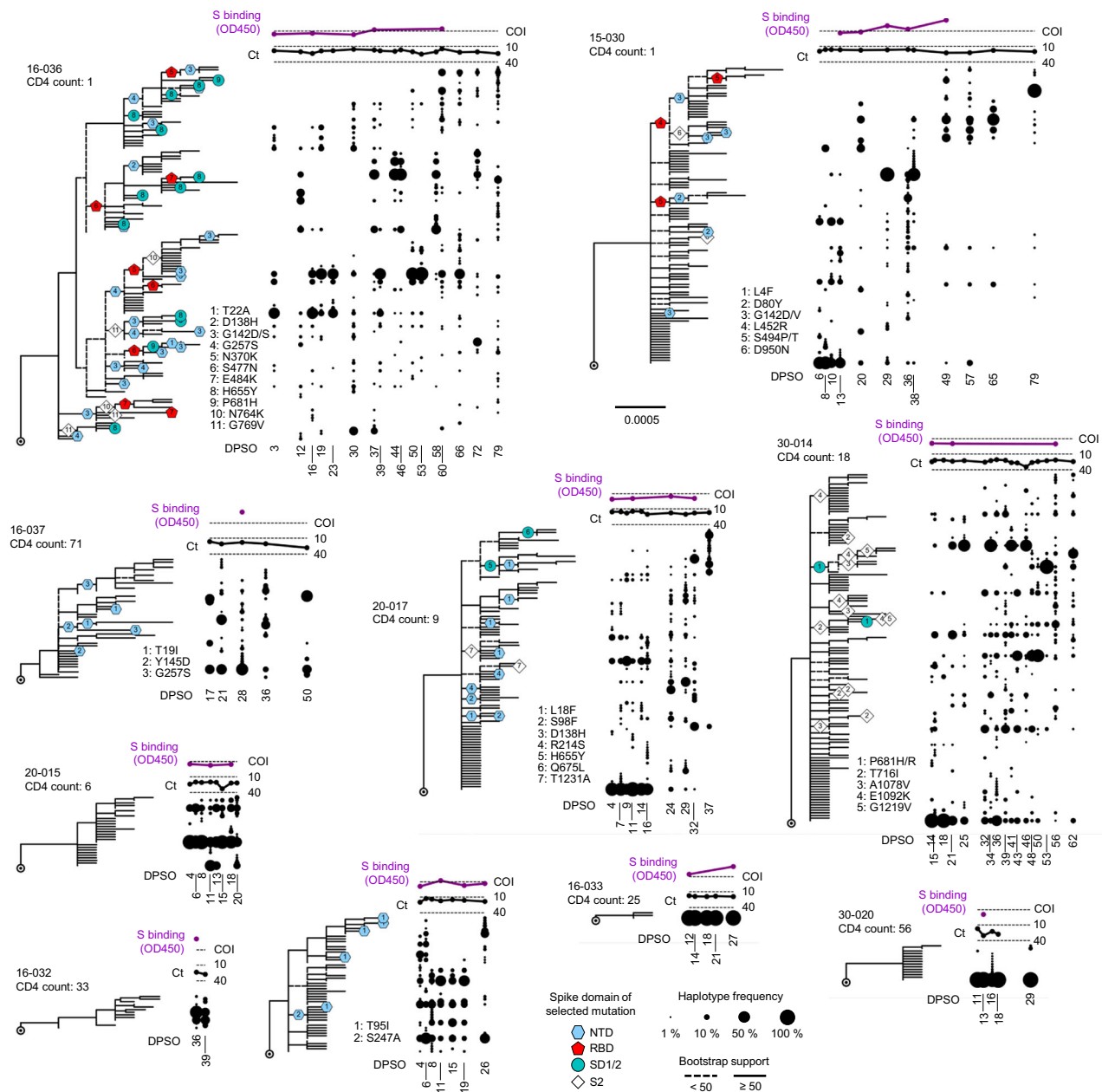

**Fig. 4 | Evolution and positive selection of SARS-CoV-2 spike in PWH with CD4 counts < 200 cells/ µL.** Maximum-likelihood phylogenetic trees rooted on Hu-1 for all haplotypes from each PWH with CD4 counts < 200 cells/µL. Participants S006-001 and S074-001 were infected with multiple founders and are shown separately in Supplementary Fig. 7a. Clades with bootstrap support < 50% are indicated with dashed lines. Sites detected under positive selection within each participant (see "Methods") are shown at their inferred location on the tree with numbered symbols; mutations corresponding to each number are listed beside each participant's tree. Symbol shapes are coded by spike protein domain (see legend, center bottom). The frequency of each haplotype detected at each sample time point (days post symptom onset) in each participant is indicated to the right of the tree via the dot plot. SARS-CoV-2 RNA levels (rRT-PCR Ct values) and serum antibody binding to spike protein (optical density, 450 nm [OD450]) are shown above the dot plot for each participant. The positivity cutoff index (COI) of 0.4 for serum antibody binding to spike protein is indicated.

changes in SARS-CoV-2 population size (as measured by Ct) were directly correlated with changes in genetic diversity (Pearson's $r = 0.30$, $p = 0.03$, see Supplementary Fig. 13). Combined with the direct correlation between with early spike diversity and subsequent SARS-CoV-2 RNA shedding duration (Fig. 2b), these findings link intra-host SARS-CoV-2 diversification and adaptive evolution to the persistence of the virus in people with advanced HIV infection.

## Discussion
Defining the extent, kinetics, and evolutionary patterns of SARS-CoV-2 diversification in individuals with immunocompromising conditions is

important for understanding both the biology of persistent infections and the emergence of new VOCs. Using specialized sequencing technology to analyze a clinically diverse cohort of PWH and PWOH, we find that permissiveness for SARS-CoV-2 replication in PWH who have low CD4 counts – often in the setting of uncontrolled HIV viremia – is associated with high levels of SARS-CoV-2 spike genetic diversity just days after COVID-19 symptom onset. Early genetic diversity in these unusual cases is likely necessary for subsequent adaptive evolution, and potentially for intra-host persistence of the infection under changing fitness constraints. Indeed, we find spike gene mutational signatures in individuals with advanced HIV infection that indicate

positive selection at sites reported to have important functional roles. Thus, SARS-CoV-2 evolution in these individuals is not solely a product of random diversification through unchecked replication but instead involves intra-host adaptation that may markedly increase the risk for the generation of new SARS-CoV-2 variants.

HIV co-infection likely promotes the intra-host persistence and evolution of SARS-CoV-2 through multiple immune insults. Of particular interest, elevated spike gene diversity detected in people with advanced HIV infection even before the expected onset of adaptive immunity could indicate innate immune defects in these individuals. Such defects may include impairment of type I interferon responses[32,33], which could allow SARS-CoV-2 to diversify rapidly through a reduced barrier to its replication[34–36]. Impaired type I interferon responses or other defects associated with comorbid chronic infection[37,38] could also promote the acquisition of multiple SARS-CoV-2 variants through a relaxed transmission bottleneck, although verifying this speculation will require studies of SARS-CoV-2 transmission pairs. Subsequently, HIV-induced defects in CD4-T-cell-dependent adaptive immunity likely compound these issues. Replicating HIV may preferentially infect activated, antigen-responsive CD4 T cells[39], and in progressive disease may also interfere with CD4 T cell help for other lymphocytes by disrupting lymph node architecture[40]. In this setting, weak, delayed, and potentially more narrowly targeted humoral immunity spike risks selecting antibody-escape mutants from a diverse variant pool. It is important to note that antiretroviral therapy (ART) may counteract innate and adaptive immune defects associated with uncontrolled HIV replication[41–44], and that previous studies have documented similar magnitude and kinetics of SARS-CoV-2-specific immune responses between PWH receiving ART and PWOH[45,46]. Therefore, while our use of specialized sequencing technology was important for describing high intra-host SARS-CoV-2 genetic diversity in this study, our striking findings likely also reflect the unique biology of advanced, uncontrolled HIV infection.

Our findings in this study have several limitations. First, although our HT-SGS technology combines high accuracy with relatively deep sampling, enzyme processivity limitations in reverse transcription (RT) and PCR prevent the efficient preparation of single-copy libraries for amplicons longer than several kilobases. While we chose to sequence spike due to a relatively high genetic diversity[47] combined with acceptable SGS yield, we acknowledge that future technical advances may enable single-copy sequencing of longer targets including informative regions outside spike. Second, mucosal swab samples like those we analyzed here cannot fully capture the cumulative virus genetic diversity present throughout the body[48,49]. Therefore, the numbers of spike sequence variants reported for each person in this study represent minimum estimates, and fluctuations in intra-host haplotype populations between samples may have been influenced by spatial variations in virus population composition. Third, because our study relied on natural infections, we are unable to determine the timing of SARS-CoV-2 transmission to our study participants. We cannot unambiguously determine which SARS-CoV-2 variants detected in each person were present at transmission, nor can we rule out that delayed symptom onset contributed to elevated SARS-CoV-2 diversity detected at early time points in some PWH. Fourth, limitations on anatomic sampling and immune analysis leave open questions about the relative importance of humoral immune pressure, selection for infectivity, and other evolutionary drivers in this setting that may be addressed in future studies using model systems with structural and functional characterization of variant sequences. Most cases of intra-host convergent evolution–i.e., detection of individual mutations in at least two phylogenetically divergent haplotypes–cannot definitively be attributed either to independent occurrence of these mutations or to intra-host recombination due to limits in the sensitivity of recombination analysis[50]. Finally, our results do not address how current Omicron subvariants might evolve in people with advanced HIV infection after prior SARS-CoV-2 infection or vaccination, nor do they establish the transmissibility between people of variants identified within each person. Further studies will be needed to understand whether intra-host SARS-CoV-2 variants arising in PWH and those arising in people with other immunocompromising conditions differ in their potential to escape pre-existing immunity in immunocompetent individuals.

Despite these limitations, our results in this study demonstrate the tremendous differences in intra-host SARS-CoV-2 genetic diversity and evolution between people with advanced, poorly controlled HIV infection and those with well-controlled or no HIV infection. We propose that these differences stem from multifactorial impairments in antiviral defenses during advanced HIV infection, which 1) enable elevated early SARS-CoV-2 replication (and, thus, SARS-CoV-2 genetic diversification) and 2) present SARS-CoV-2 with a relatively low barrier to epitope-specific mutational escape. By contrast, the virus may have less opportunity to diversify in people with well-controlled or no HIV infection, while stronger, more rapid, and potentially broader immune responses in these individuals create a relatively large barrier to escape. Regardless of the mechanism, however, the potential emergence of new pandemic virus variants in PWH who are not receiving effective ART remains highly concerning. This concern could be mitigated through active or passive immunizations that provide sufficient early protection to limit virus genetic diversification in this setting. However, our results also emphasize that efforts to control SARS-CoV-2 and potentially other viruses will benefit from addressing remaining gaps in the global approach to HIV infection.

## Methods

### Study Participants

Recruitment of study participants was performed in compliance with relevant ethical regulations. Participants provided informed consent before the study.

The hospitalized cohort was enrolled in 20 sentinel surveillance hospitals in 8 of the 9 South African provinces. Cohort enrollments were limited to individuals aged ≥18 years who were living within a 50 km radius of the respective hospitals and who had laboratory-confirmed, symptomatic COVID-19 within 5 days of diagnosis. All cohort participants underwent a combined nasopharyngeal/oropharyngeal swab at enrollment and every second day thereafter until cessation of SARS-CoV-2 shedding, as defined by 2 consecutive negative swabs. Serum specimens were collected at enrollment and days 7, 14 and 21 post symptom onset[21]. We selected individuals from the hospitalized cohort for inclusion in the present study if they had SARS-CoV-2 N gene rRT-PCR Ct ≤30 on their initial samples and at least 3 positive samples. Demographic and clinical information was collected using standardized case report forms at enrollment, daily while in hospital, and at discharge from hospital, when shedding stopped or when the individual died.

The case-ascertained household transmission study, which included the outpatient cohort, took place in Klerksdorp (North West Province) and Soweto (Gauteng Province), South Africa. Screening of index cases occurred at three clinics in Klerksdorp from October 2020 to June 2021, and at five clinics in Soweto from October 2020 to September 2021. Individuals aged ≥18 years with COVID-19-compatible symptoms starting ≤5 days before presentation were screened for SARS-CoV-2 on nasopharyngeal swabs at primary health clinics. Households of individuals positive for SARS-CoV-2 were enrolled if the index case symptoms started within 7 days before household enrollment if no other household members reported symptoms 14 days prior to household enrollment, and if there were ≥2 additional household members of whom ≥70% provided consent for the study. Households were followed for six weeks, with nasal swabs collected three times per week and serum samples collected at baseline and at the end of the follow-up period[22]. From the outpatient cohort, we

selected individuals for the present study who tested positive for SARS-CoV-2 with SARS-CoV-2 N gene rRT-PCR Ct ≤ 35. Household, demographic, and clinical information was collected in this cohort at enrollment. Information about symptoms and healthcare-seeking behavior was collected at thrice weekly follow-up visits using Research Electronic Data Capture (REDCap) databases on electronic tablets.

In the hospitalized cohort, HIV testing was conducted as part of clinical management. If an HIV diagnostic test result was not obtained during a participant's hospitalization, a prior documented positive result or evidence in hospital records of treatment with ART was considered to indicate HIV positivity. A documented negative HIV diagnostic test result within 6 months of study was considered to indicate HIV negativity. Participants with unknown HIV status or with negative HIV diagnostic test results older than 6 months were offered voluntary counseling and testing by rapid enzyme-linked immunosorbent assay (ELISA).

In the outpatient cohort, rapid HIV testing was offered for individuals with unknown HIV status, or for whom a documented negative HIV result was not available within the previous 6 months. For one participant aged 0.5 years whose mother had a documented HIV-negative status during pregnancy, HIV status was considered negative. For individuals who did not agree to rapid testing, but did consent to HIV testing, residual serum was tested for HIV antibodies by ELISA at the National Institute for Communicable Diseases (NICD). For PWH, data on CD4 counts and plasma HIV RNA levels within 6 months of enrollment were collected from medical records. If these data were not available, samples were collected and plasma HIV RNA levels tested by quantitative rRT-PCR (Roche Cobas Ampliprep/Cobas Taqman HIV-1 test, Roche Diagnostics, Mannheim, Germany) at the NICD.

For the hospitalized cohort, ethical clearance was obtained through the University of the Witwatersrand Health Research Ethics Committees (HREC) (Medical) (M160667); Stellenbosch University HREC (15206); University of Pretoria HREC (256/2020), and University of the Free State HREC (HSD2020/0625). For the outpatient cohort, clearance was obtained from the University of the Witwatersrand HREC (M2008114). Participants in the outpatient cohort received a $3.00 grocery store voucher at each follow-up visit to compensate for the time required for specimen collection and interview.

Information on biological sex was collected in these studies through self-reporting. While sex was considered as a potential exposure of interest in the design of the original studies, analysis was not stratified by sex in the current study. Out of 47 individuals enrolled in this study, 28 were female, 18 were male, and one was unknown (not reported). These details are described in Supplementary Data 1.

### Detection of SARS-CoV-2 RNA in swab specimens
Upper respiratory specimens (combined nasopharyngeal and oropharyngeal specimens in the hospitalized cohort, nasopharyngeal specimens for screening of index cases in the outpatient cohort, and nasal specimens for household follow-up in the outpatient cohort) were collected by trained study nurses using nylon flocked swabs and transported in viral or universal transport medium to the NICD for further testing. Total nucleic acids were extracted from 200 μl of each sample using the DNA/Viral NA Small Volume v2.0 extraction kit (Roche Diagnostics, Mannheim, Germany) and an automated extractor MagNA Pure 96. Detection of SARS-CoV-2 nucleic acid from specimens was performed using the Allplex™ 2019-nCoV assay (Seegene, Seoul, South Korea) with rRT-PCR. Specimens were considered positive for SARS-CoV-2 if the Ct value was < 40 for any of the E, RdRp and N SARS-CoV-2 gene targets.

### Clonal RNAs
Plasmid DNA to validate recombinant events in this study was generated by BioInnovatise, Inc. (Rockville, MD) to include the WA-1 sequence (GenBank‐ MN908947) of the 4234-bases region containing the full-

length spike gene, 310 bases of downstream sequence, and 546 bases of upstream sequence. This region was inserted into the pSI vector (Promega). A double-mutant (2 M) construct was then created by using site-directed mutagenesis to scramble 5-base regions at the 5' and 3' ends of the target (genome position 21,340 – TACAT ➝ ATTCA and genome position 25371 – TACAT ➝ ATTCA; Supplementary Fig. 4a). To prepare clonal RNA samples, plasmid constructs were transcribed in vitro using MEGAscript™ T7 Transcription Kit (ThermoFisher Scientific, AM1333) according to the manufacturer's instructions[20].

### HT-SGS
Aliquots of swab samples stored at -80°C were thawed at room temperature and centrifuged briefly before RNA extraction. Virus RNA was extracted with a magnetic bead-based RNA extraction kit (RNAdvance Viral Reagent kit, Beckman Colter, C63510) on an epMotion® 5073t liquid handler (Eppendorf, 5073000345). Sample RNA, including extracted RNA and clonal RNA, were reverse-transcribed. Procedures for RT and for purification, quantification, and PCR amplification of complementary DNA (cDNA) were as previously described[20]. RT was performed using SuperScript IV Reverse Transcriptase (ThermoFisher Scientific, 18090010) according to the manufacturer's instructions. The RT primer (synthesized by Integrated DNA Technologies [IDT]) consisted of an outer reverse primer binding site for PCR, an 8-base unique molecular identifier (UMI) generated by random base incorporation, and a gene-specific target region (CCGCTCCGTCCGACGACTCACTA-TACCCGCGTGGCCTCCTGAATTATNNNNNNNNCGTTGCAGTAGCGC-GAACAA). Note that for some validation experiments using clonal RNAs, RT primers were synthesized using pools of defined (i.e., white-listed) UMI sequences (Supplementary Data 3). After RT, cDNA was treated with proteinase K (Sigma-Aldrich, 3115828001) for 25 min at 55 °C with shaking at 1000 rpm to digest residual protein, followed by purification using RNAClean XP bead suspension (A63987, Beckman Colter) at a bead:cDNA volume ratio of 2.2:1. The cDNA copy number in a small aliquot of each sample was measured on a QIAcuity digital PCR (dPCR) system (Qiagen) using forward primer ACGTGGTGTTTAT-TACCCTGACA, reverse primer TTGGTCCCAGAGACATGTATAGC, and hydrolysis probe 5'-/56-FAM/FAM TTTCCAATGTTACTTGGTTCCA/3BHQ_1/-3' (synthesized by IDT). Cycling conditions were as follows: initial denaturation at 95°C for 2 min, followed by 45 cycles of 95°C for 15 sec and 53°C for 1 min. After dPCR quantification, an estimated 3000 cDNA templates were then subjected to full-length spike PCR. If cDNA copy numbers were not high enough to reach 3000 cDNA for sequencing, all available cDNAs were sequenced. Full-length, UMI-tagged spike gene cDNA was amplified using the Advantage 2 PCR kit (Takara Bio, 639206) with forward primer TTCGCATGGTGGACAGCCTTTGTT and reverse primer CCGCTCCGTCCGACGACTCACTATA (synthesized by IDT) under the following thermocycling conditions: initial denaturation at 95°C for 1 min; 32 cycles of 95°C for 10 sec, 64°C for 30 sec, and 68°C for 5 min; and final extension at 68°C for 10 min. PCR reagent concentrations were as follows: 800 nM forward and reverse primers, 400 μM dNTP, 1X Advantage 2 Buffer, and 2X of Advantage 2 Polymerase Mix. For long-read sequencing, amplified DNA products of length 4234 bases (encompassing the entire 3.8-kilobase spike gene) were incorporated into sequencing libraries using the SMRTbell Express Template Prep Kit 2.0 (100-938-900, Pacific Biosciences) and Barcoded Overhang Adapter kit 8 A and 8B (101-628-400 and 101-628-500, Pacific Biosciences) for multiplexing targeted sequencing. Libraries were processed through primer annealing and polymerase binding using the Sequel II Binding Kit 2.0 (101-842-900, Pacific Biosciences), and then sequenced on a Sequel II system (Pacific Biosciences) with a 20-hour movie time under circular consensus sequencing (CCS) mode.

### SGS calling
Circular consensus sequences (CCS) were generated from SMRT sequencing data with a minimum predicted accuracy of 0.99 and a

minimum of 3 passes in Pacific Biosciences SMRT Link (v11.0.0.146107)[51]. CCS reads were demultiplexed using Pacific Biosciences barcode demultiplexer (lima) to identify barcode sequences. The resulting FASTA files were reoriented into the 5'-3' direction using the vsearch −orient command in vsearch (v2.21.1). Cutadapt (v4.1) was used to trim forward and reverse primer sequences. Length filtering was performed to remove reads shorter than 3500 nt or longer than 4500 nt. Remaining reads were then binned by their 8-base UMI sequences. For each bin, reads were clustered with vsearch −cluster_fast based on 99% sequence identity. Only bins that yielded a single, predominant cluster (i.e., where the largest cluster was (1) inclusive of at least half of the bin's reads and (2) at least twice as large as the second largest cluster) with at least 10 CCS reads were kept. The cluster consensus sequence generated by vsearch −cluster_fast was then used as a reference to map the cluster's reads with minimap2 (v2.24). The commands bcftools mpileup -X pacbio-ccs and bcftools consensus were used to determine the final consensus sequence for each bin. Final consensus sequences were used as queries for BLAST nt database searches, and non-SARS-CoV-2 sequences thus identified were discarded.

Putative false UMI bins (spurious bins that arise due to PCR and/or sequencing errors) were identified and removed with a network approach as previously described[20]. Given two distinct bins a and b with read counts $n_a$ and $n_b$, and assuming $n_a \geq n_b$, a and b are connected by an edge if they have edit distance 1 and satisfy the following count criterion: $n_a \geq 2n_b - 1$. Networks formed as above were resolved using the adjacency method[52], which iteratively consolidates smaller bins into larger bins that meet the above criteria. As a final filter, a mixture model of bin size was iteratively optimized using exponential and Gaussian distributions representing false and real bins, respectively. Bins with posterior probability Prob(false) > 0.5 were discarded and the remaining bins were used as final SGS. The full HT-SGS data processing pipeline used is available at https://github.com/niaid/UMI-pacbio-pipeline/releases.

## Variant and haplotype calling

Despite high CCS read accuracy and UMI-based error correction, sample RT errors and other rare errors nonetheless persist in processed HT-SGS datasets. To address such errors, variant calling was performed using a model describing technical error rates. Given an RT error rate $R = 1 \times 10^{-4}$, a target insert length L, and a number of recovered SGS sequences N, the probability of observing a technical variant with at least c occurrences in the sample was expressed as $P(C \geq c) = 1 - \text{Binom}_{CDF}(c \,|\, N, R)^L$. To determine a cutoff for variant calling, the smallest value $c_v$ for which $P(C \geq c_v) < \alpha$ was determined, where $\alpha = 0.01$ is the minimum significance threshold. The minimum number of occurrences to call a variant was set as $c_v + 1$. Indels were handled separately; at least three identical occurrences of an indel were criteria for inclusion as a real variant. Variants in each sample not meeting these criteria were reverted to the consensus of all SGS for that sample. This variant calling approach was implemented with a custom Python script that is available at https://github.com/niaid/UMI-pacbio-pipeline/releases. Among SGS subjected to variant calling, each unique combination of mutations within the individual was considered as one haplotype. To avoid inaccurate findings arising from any residual erroneous sequences, only SGS representing haplotypes that were detected at least 2 times in each sample were included in downstream analysis.

## HT-SGS with whitelisted UMIs

For experiments using whitelisted UMIs, UMI sequences were designed using DNABarcodes[53] enforcing a minimum Levenshtein distance of 4, setting filter.self_complementary as "true," and setting the search heuristic as "conway." The complete pool of whitelisted UMI primer sequences used is available in Supplementary Data 3. Whitelisted UMI primer pools were used via the same protocol used for 8 N (random) UMI pools as described above. When calling SGS from CCS reads, only bins with UMI sequences matching UMIs from the whitelisted pool were considered.

## Resampling of SGS for estimation of variant false discovery rates

To increase the size of our validation dataset, we randomly resampled sequences from the aligned SGS obtained from sequencing of the wild-type (WT) and mutant spike constructs with the 'random' module of Python. Resampled datasets were obtained at a depth of 10, 50, 100, 500, and 1000 SGS. One hundred (100) replicates at each level were performed for WT and mutant SGS datasets, yielding 200 total "samples" at each level. These samples were then processed with our variant and haplotype calling approach as described.

## Identification of technical recombinant sequences

To identify recombinant sequences arising in control experiments using WT and 2 M spike constructs (Supplementary Fig. 4), the program cutadapt (v 4.1) was used with the following settings to label reads according to their anchor sequences; left anchor: leftWT= CATGCATGCAAATTACATATTTTGGAGGAATACA, left2M= CATGCATGCAAATATTCAATTTTGGAGGAATACA, minimum overlap of 33 bases, maximum error count of 3; right anchor: rightWT= AAAGGAGTCAAATTACATTACACAT, right2M= AAAGGAGTCAAATATTCATACACAT, minimum overlap of 24 bases, maximum error count of 3. To account for the fact that recombination is only detectable when two different sequences recombine (i.e., recombination between identical sequences will not be detected), we estimated the total recombination rate as 2 times the observed recombination rate.

## Normalized Shannon entropy

Normalized Shannon Entropy ($H_{norm}$) for a group of aligned sequences was calculated as the entropy for those sequences (H) divided by the maximum possible entropy for that number of sequences ($H_{max}$), $H_{norm} = H/H_{max} = -\sum_{i=1}^{n} p(x_i) \log_2(p(x_i))/\log_2(n)$.

## Average pairwise genetic distance

The average pairwise distance for each group of sequences was calculated as the total number of mutations (point mutations and/or indels) between each pair of sequences divided by the number of pairs in that group.

## Jensen-Shannon distance

Within each participant, spike sequence population dissimilarity (distance) was analyzed for all possible sample pairs. The Jensen-Shannon calculation was performed for each sample pair using the haplotype frequency distributions for the samples and the "Jensen-Shannon" method in SciPy (v 1.8.1).

## Estimation of mutations below detection limit

We anticipated that some real variant sequences might be inadvertently removed by our data analysis process due to sampling depth limitations. To estimate the number of uncalled biological mutations (MU) in each sample after analysis, we considered the total number of mutations in the sample before variant calling (M0), the number of mutations that exceeded the variant calling threshold and were therefore interpreted as real (M1), and an estimate of the number of technical mutations (e.g., RT errors) expected in the sample (M2; based on an assumption of $1 \times 10^{-4}$ error/base). M0 and M1 were counted directly from an alignment of all SGS for the sample, and M2 was computed as the upper 99% CI of a Binomial distribution with N as the total number of nucleotides in the alignment and $p = 1 \times 10^{-4}$, as computed via SciPy (v 1.8.1). MU was then calculated as $MU = M0 - M1 - M2$ and restricted to be zero or greater. We considered values of MU > 0 to imply the presence of real mutations that went uncalled after the variant and haplotype calling process.

## Phylogenetic inferences

Recombinant sequences were identified using 3SEQ[54] (v 1.8.0) using "full run" mode on each participant's haplotypes and a threshold of $p < 0.05$. Recombinant sequences thus identified were excluded from initial phylogenetic models and analyses of positive selection.

Phylogenetic relationships were inferred within each participant using all non-recombinant haplotype sequences for that participant and with Wuhan-Hu-1 spike (GenBank Accession NC_045512.2, nucleotide coordinates 21563-25384) included as an outgroup. To account for indels when performing phylogenetic analyses, a binary matrix was generated using 2matrix[55], which encoded for the presence or absence of indels in each haplotype. This matrix was computed using a curated combined alignment of all haplotypes across all participants. Trees were then constructed using iqtree[56] (v 1.6.12) with a partitioned model. Nucleotide sequences were analyzed using an HKY model, while the indel matrix was analyzed using a JC2 morphological model with enforced transition rates of 0.99 (indel acquisition) and 0.01 (indel reversion). Maximum-likelihood trees were computed, and support values were obtained through ultra-fast bootstrapping with 1000 iterations.

The presence of multiple founder sequences was assessed using TreeCluster[25]. Clusters were formed using the "single linkage" mode of TreeCluster with a phylogenetic distance threshold of 0.0015, which corresponds to > 5 mutations within the 3822 nt coding sequence of the spike. For participants in whom multiple clusters were detected, each cluster was re-processed with the phylogenetic method described above to yield subtrees corresponding to each founder; recombinant sequences identified earlier were added to the associated cluster most divergent from Wuhan-Hu-1. The subtrees were concatenated into a single tree with a custom Python script, and then branch lengths were reoptimized with iqtree.

## Testing for selection

Testing for positive selection was performed initially with FUBAR[29] using trees of participant haplotypes computed as described above. Outlying sequences in each participant with less than 80% amino acid homology to other sequences were excluded before analysis, thereby removing most premature stop codons, large deletions, and frameshifts. For $\omega$ equal to the relative rate of nonsynonymous over synonymous mutations at a site, we considered sites with $\text{Prob}(\omega > 1) > 0.9$ to be under possible positive selection. To confirm positive selection for these sites, we examined frequencies of associated nonsynonymous mutations in the participants over time. Frequency changes were assessed relative to the participant's first sample and considered the summed frequencies of all haplotypes containing a given mutation. Mutations with a frequency increase of at least 0.20 in the participant and $\text{Prob}(\omega > 1) > 0.9$ via FUBAR were called positive selection.

## Short-read WGS data

Short-read data from selected samples from both the hospitalized and outpatient cohorts were obtained with the Ion Torrent Genexus platform using AmpliSeq for SARS-CoV-2[21,22], following the ARTIC SARS-CoV-2 sequencing protocol (https://artic.network/ncov-2019). Data were processed with a unified pipeline in CLC Genomics Workbench v22.0.3. This pipeline included quality filtering, read trimming, mapping to the Wuhan-Hu-1 reference (Hu-1, GenBank Accession NC_045512.2), local realignment, consensus calling, and variant calling. Variant calling was performed with a minimum coverage of 10, minimum variant count of 2, and minimum variant frequency of 1%.

## Measurement of spike antibody binding responses

In both the hospitalized and outpatient cohorts, antibodies against the SARS-CoV-2 spike protein were detected using an enzyme-linked immunosorbent assay (ELISA) as previously described[57].

Recombinant trimeric spike protein (source: South African Medical Research Council Antibody Immunity Research Unit [SAMRC AIRU]) was coated onto 96-well, high-binding plates at a concentration of 2 µg/ml and incubated overnight at 4 °C. Subsequently, the plates were washed and blocked using a blocking buffer containing 5% skimmed milk powder, 0.05% Tween 20, and 1× PBS, followed by incubation at 37 °C for 1–2 hours. Serum samples diluted 1:100 and control antibodies (positive: CR3022 [SAMRC AIRU] and negative: Palivizumab [Medimmune, RRID: AB_2459638]) diluted to 10 µg/mL were added to the plates, followed by a one-hour incubation at 37 °C. Next, an anti-human horseradish peroxidase-conjugated antibody (Merck, catalog # A0170-1) was added and incubated for another one hour at 37 °C. To visualize the antibody binding, a OneStep TMB substrate (Thermo Fisher Scientific, USA) was added and allowed to develop for 5 min at room temperature. The reaction was stopped by adding 1 M $H_2SO_4$ stop solution. The absorbance at 450 nm was measured, and specimens with optical density (OD) > 0.4 were considered positive for anti-spike antibodies.

## Solvent accessible surface area (SASA) in the spike trimer

Solvent accessible surface area was assessed via NACCESS[27] on an atomistic structure model of the spike trimer generated using YASARA (htttp://www.yasara.org) on a D614G structure template (PDB: 7KRQ)[28]. The accessibility of each residue was computed as the mean of the 3 accessibilities of the residue over the 3 protomers in the trimer.

## Comparison of spike mutations with public data

Publicly available SARS-CoV-2 spike nucleotide sequences were downloaded from the NCBI Virus database (https://www.ncbi.nlm.nih.gov/labs/virus/vssi/#/), filtering for the South Africa region and collection dates between April and October 2020. The spike sequences were extracted from the downloaded sequences and aligned to Hu-1, and the number of relative point mutations was counted for each sequence. This process was repeated for each intra-host SGS from within the collection period.

## Analysis of the intra-host spike sequences with Nextclade

All intra-host sequences were uploaded to Nextclade web (https://clades.nextstrain.org). The results were downloaded in the next clade.tsv file and the Nextclade_pango lineage field was extracted for each of the input sequences. The relative frequency of each assigned Nextclade Pango lineage per host was used to plot the frequencies of intra-host Pango lineages.

## Statistical analyses

GraphPad Prism v.9.3.1 and MATLAB 2022a were used for statistical analyses. Specific statistical tests used in each analysis are presented in the corresponding figure legend. The significance of single comparisons in multiple groups was assessed by one-way ANOVA with multiple comparisons using the Kruskal-Wallis test (unpaired or unmatched groups) or Friedman test (paired groups) and Dunn's multiple comparisons test. The nonparametric Spearman's rank correlation test (two-tailed), Pearson correlation coefficient (two-tailed), and simple linear regression were used for correlation analyses. Modeling of the primary major haplotype frequency was performed with logistic regression via the method of iteratively reweighted least squares.

## Reporting summary

Further information on research design is available in the Nature Portfolio Reporting Summary linked to this article.

## Data availability

All data that support the findings of this study are available in this article and supplementary materials. Long-read sequencing data generated in this study have been deposited in the NCBI SRA database with

accession codes SRR27325889–SRR27326072 under BioProject PRJNA1055920. Source data are provided with this paper.

## Code availability

UMI-pacbio-pipeline v.1.1 was used to generate single-genome sequences and call haplotypes. This pipeline is available at https://github.com/niaid/UMI-pacbio-pipeline. Code used to generate data supporting the findings of this study are available with processed data (single-genome sequences, haplotypes, and phylogenetic trees) at Zenodo (https://zenodo.org/doi/10.5281/zenodo.12744612)[58].

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

## Acknowledgements

We gratefully acknowledge the participants in this study. We thank Sibongile Walaza for assistance with the clinical studies. This work was supported by the NIH Intramural Research Program (project number AI005157-01, to EAB), and by the Wellcome Trust (grant 221003/Z/20/Z, to CC) in collaboration with the Foreign, Commonwealth and Development Office, United Kingdom, the US Centers for Disease Control and Prevention (co-operative agreement 6 U01IP00104804-02, to CC), as well as the National Institute for Communicable Diseases, a division of the National Health Laboratory Service, South Africa (to CC and AVG).

## Author contributions

S.H.K., J.N.B., A.V.G., C.C., and E.A.B. conceived the study. J.N.B., S.M., J.K., D.A., D.K., N.M., L.L., J.E., S.T., N.W., A.V.G., and C.C. were responsible for clinical sample acquisition and primary testing. J.N.B., S.M., J.K., N.M., L.L., S.T., N.W., A.V.G., and C.C. managed the clinical sample acquisition study. S.H.K. and M.L. performed HT-SGS of SARS-CoV-2 spike genes. P.R. and F.B. performed bioinformatic analysis. P.R., F.B., and V.G.C. performed phylogenetic and evolutionary analysis. D.A., J.N.B., D.K., and N.W. performed short-read, whole-genome sequencing of SARS-CoV-2. T.B., R.R., and P.D.K. performed solvent-accessible surface area (SASA) analysis. A.V.G., C.C., and E.A.B. obtained/provided resources for the study. S.H.K., P.R., F.B., and E.A.B. wrote the manuscript. All authors edited the manuscript. A.V.G., C.C., and E.A.B. supervised the study.

## Funding

## Competing interests

CC has received funding from Sanofi Pasteur, the Bill and Melinda Gates Foundation, US Centers for Disease Control and Prevention (CDC), South African Medical Research Council and Wellcome Trust to support research unrelated to this study. AVG and NW have received grant funding from the United States Centers for Disease Control, the Bill and Melinda Gates Foundation, and Sanofi Pasteur to support research unrelated to this study. NM discloses institutional funding from Pfizer for a separate study of patients with pneumonia. The remaining authors declare no competing interests.

## Additional information

[1]Vaccine Research Center, National Institute of Allergy and Infectious Diseases, National Institutes of Health, Bethesda, MD, USA. [2]National Institute for Communicable Diseases, a division of the National Health Laboratory Service, Johannesburg, South Africa. [3]SAMRC Antibody Immunity Research Unit, Faculty of Health Sciences, University of the Witwatersrand, Johannesburg, South Africa. [4]School of Public Health, Faculty of Health Sciences, University of the Witwatersrand, Johannesburg, South Africa. [5]Department of Integrative Biology and Bioinformatics, College of Biological Sciences, University of Guelph, Ontario, Canada. [6]Perinatal HIV Research Unit, University of the Witwatersrand, Johannesburg, South Africa. [7]Johns Hopkins University, Center for TB Research, Baltimore, MD, USA. [8]School of Pathology, Faculty of Health Sciences, University of the Witwatersrand, Johannesburg, South Africa. [9]These authors contributed equally: Sung Hee Ko, Pierce Radecki. [10]These authors jointly supervised this work: Cheryl Cohen, Eli A. Boritz. ✉e-mail: eli.boritz@niaid.nih.gov

