## [Peer Review File · Nature Communications]

Rapid Intra-host Diversification and Evolution of SARS-CoV-2 in Advanced HIV InfectionReviewers' Comments:

Reviewer #1:

Remarks to the Author:

Comments

The authors employ HT-SGS to elucidate the intricate evolutionary processes of SARS-CoV-2 within hosts, categorizing them into PWOH, advanced PWH, and non-advanced PWH. In the cases with lower immune response to SARS-CoV-2 (PWOH and PWH with CD4 counts ≥ 200 cells/L), the study found there is little haplotype diversity within host. While advanced HIV infection shows heightened diversity and frequent replacement with main haplotypes (approximately 46 haplotypes per person), suggesting positive selection for potential VOC emergence.

Several studies corroborate these findings, showcasing increased genetic diversity in persistently infected individuals, such as those coinfecting with HIV or immunocompromised due to chemotherapy and/or lymphoma. The study's novelty, as the authors claimed, lies in the application of HT-SGS (incorporating UMI) to trace detailed processes, especially very low-frequency haplotypes, within hosts at the haplotype level.

To fortify the claim of a more comprehensive understanding of within-host evolutionary processes, an inquiry arises regarding the decision to sequence only the Spike genes and not the whole genome (~30 kb). Technologies such as PacBio HiFi and ONT ultra-long reads offer read lengths exceeding 10 kb or even 100 kb. Therefore, why did the study opt not to sequence the entire genome, and what potential obstacles were considered?

The authors claimed that SARS-CoV-2 RNA levels (which could reflect the number of virus particles within host, i.e., the population size of virus within host) were largely independent of diversity measures within subgroups (Extended Data Fig. 3d), which seems unexpected. The larger population size resulted from elevated cumulative numbers of replicative cycles and more mutations will emerge. Thus, the pairwise distance between any viral particles is supposed to increase with the higher population size. Many independent/dependent possible forces, e.g., RNA levels (population size), SGS count, CD4 level, shedding duration, and so on, could influence the genetic diversity. The authors could try some analysis, like multiple linear regression, to disentangle their respective contributions to genetic diversity.

The observed adaptive selection and higher diversity in PWH with advanced infection pose a paradox. Traditional evolutionary theory predicts a reduction in genetic variation under adaptive selection due to selection sweeps favoring beneficial mutations. The authors could provide insights into this apparent contradiction to reconcile the observed results with established population genetics principles.

The recurrent mutations across different clades in the phylogenetic trees (Fig. 4) prompt questions about their origin. Clarification on whether these mutations independently occurred multiple times within a single host or resulted from recombination events would enhance the interpretation. It will

be much better to see a supplementary figure depicting the frequencies of each haplotype within each host over time, perhaps like Muller Diagram. Given there are several timepoints, I imagine this would look a bit messy, but what I'm interested in is to what extent a few haplotypes take over entirely.

In summary, while the heightened genetic diversity in patients with advanced HIV infection aligns with expectations, the application of HT-SGS enriches our understanding of within-host evolutionary dynamics. The high-resolution insights at the haplotype level provide direct evidence supporting recombination events, a challenging detection using traditional consensus sequencing.

Minor comments/suggestions, noted by Page/Line.

1. P6/L127. "SARS-CoV-2 RNA levels were largely independent of diversity measures within subgroups (Extended Data Fig. 3d)." Upon initial review of Extended Data Fig. 3d, it appears that RNA levels (negatively correlated with Ct value) are positively correlated with genetic diversity, especially in the middle panel. Providing linear regression information would clarify this apparent discrepancy.

2. P12/L218. "We performed single-linkage phylogenetic clustering to identify clades of 218 haplotypes in each person that were separated by at least 5 mutations and were thus likely to have originated from distinct founder". While the choice of 5 mutations is suitable for distinguishing different founder haplotypes, a comparison of donor-recipient haplotypes could offer a more accurate estimation of transmission founder size. Is there any available epidemiological information on transmission pairs among the patients in this study?

3. Fig. 1c y-axis label. "Number of mutation relative Hu-1". Add the preposition "TO" for clarity: "Number of mutations relative to Hu-1."

4. Extended Data Fig. 4. The summary frequency of all possible haplotypes is expected to be approximately 1 for each sample. However, in several samples (e.g., 20-017, 20-015, 16-032), the summary frequency does not exceed 80%. Could this be attributed to numerous haplotypes with very low frequencies? Clarification on this aspect would be helpful.

5. Fig. 3 legend. "Fig.3. Analysis of intra-host spike mutations in PWH and PWOH." A missing space in "Fig.3". Correcting it to "Fig. 3" for consistency.

Reviewer #2:

Remarks to the Author:

It has been known that SARS-CoV-2 can accumulate more mutations in people with HIV (PWH). In this study, Ko et al tried to understand the underline mechanisms by studying S sequences obtained by HT-SGS between PWH and PWOH populations. They show that haplotypes are more

frequently detected in PWH than PWOH. However, there are some major concerns that need to be carefully addressed.

The HT-SGS is not likely the real “SGS”. It is not clearly described in this study and the previous publication (I may have missed if it was) how the first round PCR was done. If it was carried as with bulk PCR (all templates together), there will be extensive PCR-derived artificial recombinants during amplification. This is well known in the field and such artificial recombinants can account for up to 60% of the population. This will be especially worse for the amplicon as long as 4.3kb in this study. UMI will not resolve this problem. That is why the real SGS method has been developed by Coffin’s group also at NIH (PMID: 15635002) and widely used by others. At least the authors should use the Coffin SGS method to confirm their findings.

The high numbers of haplotypes at the very early stage (3 or 4 DPSO) and persistent but similar levels of haplotypes throughout the studied period in PWH strongly indicate that these haplotypes were most likely the results of PCR-derived artificial recombinants during amplification. In addition, the haplotype profiles change dramatically just 1-2 days apart (Fig 4). Viral population should not change that fast.

They have obtained 70,968 sequences from 184 samples from 47 cases. However, it is not clear how many sequences are obtained for each time point from each case. How haplotypes are defined? Is one S gene with one mutation considered one haplotype? At what percentage a haplotype should be considered reliable?

Do the phylogenetic trees include all the sequences from each case in Fig. 4? If so, are there identical sequences in each case? The sequences from different time points should be clearly indicated in the tree. Do viruses, not haplotypes, evolve over time? What are the diversity levels from different time points in each case? Does it change over time? These are the key information that should be presented.

The all groups are not comparable. The time for the PWOH, PWH (CD4 \geq 200) and PWH (CD4 unknow) were very short, while PWH (CD4 <200) were followed much longer time. It is very likely that the time is the determining factor. Is the sequence for each case in Fig. 3 the accumulation all detectable mutations? If so, this may be misleading since some very rare ones will carry the same weight as those predominant ones.

It is surprising to see high level variability only 3 or 4 DPSO in PWH. What caused this? This cannot be explained by poor immunity in PWH. SARS-CoV-2 does not change much in cell culture.

The abstract is misleading. The study did not really show “rapid emergency and evolution of SARS-CoV-2”

Other regions should also be studied. At least one region from a few cases.

The R values are too low to be meaningful in Fig. 2.

At what levels, two founders were defined?

Reviewer #3:

Remarks to the Author:

This study focuses on the within-host evolutionary kinetics of SARS-CoV-2 in people with and without HIV (controlled and poorly controlled). The study finds that in people with poorly controlled HIV, SARS-CoV-2 evolution is marked by rapid diversification by the time of symptom onset and higher diversity of the within-host virus population as compared to people without HIV. The manuscript is very well written and was a joy to read. The approach, methods, and conclusions are all clear and well justified.

My sole critique relates to the first paragraph of the introduction - Line 55 – “However, convergent evolution of the same mutations in unrelated persistent cases implies extensive early intra-host SARS-CoV-2 sequence diversification that has not been directly observed. Many persistent infections described in previous studies have been characterized retrospectively, with limited analysis during the acute phase.” – This feels like it is being put forth as the unique selling point for this story but I don’t follow the logic here. This could also be the case if there are limited high fitness evolutionary paths. And capturing the acute phase is fine, but the real strength of this study is the number of study participants allowing for more robust conclusions and a more authoritative presentation of results. I suggest highlighting this point.

Reviewer #1:

Summary: The authors employ HT-SGS to elucidate the intricate evolutionary processes of SARS-CoV-2 within hosts, categorizing them into PWOH, advanced PWH, and non-advanced PWH. In the cases with lower immune response to SARS-CoV-2 (PWOH and PWH with CD4 counts ≥ 200 cells/ μ L), the study found there is little haplotype diversity within host. While advanced HIV infection shows heightened diversity and frequent replacement with main haplotypes (approximately 46 haplotypes per person), suggesting positive selection for potential VOC emergence.

Several studies corroborate these findings, showcasing increased genetic diversity in persistently infected individuals, such as those coinfecting with HIV or immunocompromised due to chemotherapy and/or lymphoma. The study's novelty, as the authors claimed, lies in the application of HT-SGS (incorporating UMI) to trace detailed processes, especially very low-frequency haplotypes, within hosts at the haplotype level.

In summary, while the heightened genetic diversity in patients with advanced HIV infection aligns with expectations, the application of HT-SGS enriches our understanding of within-host evolutionary dynamics. The high-resolution insights at the haplotype level provide direct evidence supporting recombination events, a challenging detection using traditional consensus sequencing.

We thank the reviewer for these positive comments about the study.

Major comments

Comment 1. To fortify the claim of a more comprehensive understanding of within-host evolutionary processes, an inquiry arises regarding the decision to sequence only the Spike genes and not the whole genome (~30 kb). Technologies such as PacBio HiFi and ONT ultra-long reads offer read lengths exceeding 10 kb or even 100 kb. Therefore, why did the study opt not to sequence the entire genome, and what potential obstacles were considered?

Author reply: We thank the reviewer for this important clarifying question. We do agree that the ideal sequencing method would yield highly-accurate, high-throughput, *single-copy* data for full-length SARS-CoV-2 genomes (i.e., ~30 kb), and we acknowledge the 10-100 kb read lengths of PacBio HiFi and ONT. However, the UMI-tagged amplicon library construction in our HT-

SGS approach requires that the entire target region be reversed-transcribed and PCR-amplified in a single fragment. Commercially-available reverse-transcriptases and PCR polymerases do not efficiently recover fragments of more than several kilobases in this process. These considerations are partially addressed in our initial publication from 2021 [1], which showed that SGS yield loss for cDNA corresponding a 7-kb region including S, E, and M genes was approximately 87% (see Table 2 in ref [1]). Importantly, this reflects only the PCR amplification and downstream steps of the process, and does not reflect losses due to sample fragmentation and limited reverse-transcriptase processivity. Based on these considerations, we believe that recovery of a ~30 kb amplicon in our approach would have been near zero.

Notwithstanding technical reasons for our decision to sequence only spike in this study, we agree with the reviewer that the manuscript should clearly address our rationale for sequencing target selection. In this regard, we note that intra-host and inter-host diversity of SARS-CoV-2 is particularly high in spike [3], making this region highly sensitive for distinctions between closely related haplotypes. We have now clarified these thought processes in the revised manuscript as follows (lines 384-389 in Discussion section).

“Our findings in this study have several limitations. First, although our HT-SGS technology combines high accuracy with relatively deep sampling, enzyme processivity limitations in reverse-transcription (RT) and PCR prevent the efficient preparation of single-copy libraries for amplicons longer than several kilobases. While we chose to sequence spike due to a relatively high genetic diversity⁴⁷ combined with acceptable SGS yield, we acknowledge that future technical advances may enable single-copy sequencing of longer targets including informative regions outside spike.”

Reference for the Comment 1

1. S. Ko et al., “High-throughput, single-copy sequencing reveals SARS-CoV-2 spike variants coincident with mounting humoral immunity during acute COVID-19,” *PLoS Pathog* **17**, e1009431 (2021).
2. R.E. Workman et al., “Nanopore native RNA sequencing of a human poly A transcriptome.” *Nat. Methods* **16**, 1297-1305 (2019).

3. M. Amicone et al., “Mutation rate of SARS-CoV-2 and emergence of mutators during experimental evolution,” *Evol. Med. Public Health* **10**, 142-155 (2022).

Comment 2. The authors claimed that SARS-CoV-2 RNA levels (which could reflect the number of virus particles within host, i.e., the population size of virus within host) were largely independent of diversity measures within subgroups (Extended Data Fig. 3d), which seems unexpected. The larger population size resulted from elevated cumulative numbers of replicative cycles and more mutations will emerge. Thus, the pairwise distance between any viral particles is supposed to increase with the higher population size. Many independent/dependent possible forces, e.g., RNA levels (population size), SGS count, CD4 level, shedding duration, and so on, could influence the genetic diversity. The authors could try some analysis, like multiple linear regression, to disentangle their respective contributions to genetic diversity.

Author reply: We appreciate this question, which points out an area in which our original manuscript was unclear. As the reviewer suggests, the SARS-CoV-2 RNA level in each sample likely reflects the number of SARS-CoV-2 replicative cycles in the person leading up to the time of sampling. Replication creates new variants that can expand in the person by drift or selection. Some SARS-CoV-2 variants may expand sufficiently to be detected at the depth of sampling achieved by our HT-SGS method, contributing to measured diversity.

Nonetheless, we believe that the lack of a clear correlation between SARS-CoV-2 RNA levels and genetic diversity in our original Extended Data Fig. 3d (Supplementary Fig. 5 in the revised manuscript) is to be expected. First, as the reviewer notes, many independent/dependent possible forces could influence genetic diversity. Although SARS-CoV-2 RNA levels measured at a given timepoint during infection may correlate with the number of replicative cycles leading up to that measurement, RNA levels also peak early in infection [1], before the onset of adaptive immune responses that select for the expansion of virus sequence variants. Thus, immune selection for virus sequence diversity may tend to create an *inverse* correlation between diversity and SARS-CoV-2 RNA levels over the course of each individual’s infection. Second, because SARS-CoV-2 RNA levels often decline over the course of infection, they do not always directly reflect the

cumulative numbers of replicative cycles that have occurred in the person. In other words, during each individual's infectious course, some of the lowest SARS-CoV-2 RNA levels may occur at the very end, after the largest cumulative numbers of replicative cycles have occurred (and adaptive immune responses have been mounted). The revised manuscript Discussion now touches on some of these speculative considerations.

In view of this complexity, we feel that the correlation analysis presented in the original Extended Data Fig. 3d (Supplementary Fig. 5 in the revised manuscript) risks oversimplification and confusion. Therefore, we have excluded this analysis from the revised manuscript, and thank the reviewer for calling attention to the issue.

Reference for the Comment 2

1. O. Puhach et al., "SARS-CoV-2 viral load and shedding kinetics," *Nat Rev Microbiol* **21**, 147-161 (2022).

Comment 3. The observed adaptive selection and higher diversity in PWH with advanced infection pose a paradox. Traditional evolutionary theory predicts a reduction in genetic variation under adaptive selection due to selection sweeps favoring beneficial mutations. The authors could provide insights into this apparent contradiction to reconcile the observed results with established population genetics principles.

Author reply: We agree with the reviewer that our initial submission did not do enough to reconcile our results with established populations genetics principles regarding the expected relationship between adaptive selection and high diversity. As the reviewer indicates, traditional evolutionary theory predicts a reduction in genetic variation under adaptive selection. Our findings in PWH with CD4 counts <200 cells/ μ L are consistent with this prediction. Two examples of this are the emergence of P681H after day 50 in participant 30-014 and the emergence of L452R-containing haplotypes after day 36 in participant 15-030, which are now visualized in Muller diagrams according to the reviewer's suggestion (see below and Supplementary Fig. 12). We have also revised the text in the section "Selection and autologous antibody responses," lines 324-327 and 331-334 to highlight these examples in greater detail.

“Increased frequency of L452R in participant 15-030 was associated with reduced SARS-CoV-2 RNA levels and a decrease in population diversity (i.e., entropy), followed by increased SARS-CoV-2 RNA levels and continued spike gene diversification (Supplementary Fig. 12) [...] Notably, P681H was detected in 30-014 after an apparent population bottleneck (i.e., reduced SARS-CoV-2 RNA levels as reflected by Ct >35) and was followed by increased virus load and emergence of new spike variants (Supplementary Fig. 12)”

More generally, we found among PWH with CD4 counts < 200 cells/ μ L that reductions in population size (reflected by increasing Ct value) significantly predicted reductions in diversity (as measured by entropy), as expected for selection sweeps. This has now been added to the manuscript as Supplementary Fig. 13 and discussed in the section “Selection and autologous antibody responses,” lines 340-343, as below.

“Considering all pairs of successive timepoints in PWH with CD4 counts <200 cells/ μ L, we found that changes in SARS-CoV-2 population size (as measured by Ct) were directly correlated with changes in genetic diversity (Pearson’s $r = 0.30$, $p = 0.03$, see **Supplementary Fig. 13**).”

We also note that the reviewer’s comment raises an excellent question about selection and diversity in PWH with CD4 counts <200 cells/ μ L, as compared to other groups. Considering that intra-host positive selection is identified almost exclusively among PWH with CD4 counts <200 cells/ μ L, why is diversity **higher** among these participants than among those in the other groups? We believe here again that the complexity of the biology requires more explanation that was provided in our initial submission. Very likely, lower levels of SARS-CoV-2 replication (reflected by lower RNA levels and numbers of synonymous mutations) may be relatively limiting for the generation of intra-host diversity in people with healthy immune systems, as compared to those with advanced HIV. Compounding this, combinations of mutations that the virus requires to escape intra-host from relatively strong, broadly-targeted antiviral responses in immunocompetent people may be extensive. Thus, SARS-CoV-2 may have a relatively limited replicative opportunity to generate diversity in immunocompetent people, compounded by a relatively large genetic barrier to escape. We have now discussed these considerations in more detail in the manuscript Discussion section (lines 413-419), and thank the reviewer for calling attention to this central issue.

“We propose that these differences stem from antiviral defense impairments during advanced HIV infection that promote early SARS-CoV-2 genetic diversity and enable subsequent SARS-CoV-2 adaptive evolution. By contrast, SARS-CoV-2 may have less opportunity to diversify in people with well-controlled or no HIV infection, while stronger, more rapid, and potentially broader immune responses in these individuals create a relatively large genetic barrier to SARS-CoV-2 mutational escape.”

Comment 4. The recurrent mutations across different clades in the phylogenetic trees (Fig. 4) prompt questions about their origin. Clarification on whether these mutations independently occurred multiple times within a single host or resulted from recombination events would enhance the interpretation. It will be much better to see a supplementary figure depicting the frequencies of each haplotype within each host over time, perhaps like Muller Diagram. Given there are several timepoints, I imagine this would look a bit messy, but what I'm interested in is to what extent a few haplotypes take over entirely.

Author reply:

We thank the reviewer for the helpful suggestion to show the haplotype frequencies in each host over time in a visualization like a Muller Diagram. We have now included this visualization for all participants with 2 or more timepoints sequenced as Supplementary Fig. 12.

With regards to whether recurrent mutations were independent or resulted from recombination, we found that quantitative 3seq analysis identified recombinant sequences only in participant S074-001, who experienced a dual infection with Delta and C.1.2 variants. The dual infection with genetically divergent variants was likely critical for allowing detection of recombination in this case. Accordingly, it remains possible that recombination occurred in other participants, but that the parental haplotypes were too similar for the recombination to be identified. This is now acknowledged explicitly in the revised manuscript, in the two quoted sections below (lines 228-229 in 'Analysis of transmitted SARS-CoV-2 spike diversity' section and lines 400-404 in Discussion section)

“Recombinant haplotypes were not detected in the other participants, potentially due to limited sensitivity of recombination analysis in the setting of low intra-host genetic diversity.”

“Most cases of intra-host convergent evolution – i.e., detection of individual mutations in at least two phylogenetically divergent haplotypes – cannot definitively be attributed either to independent occurrence of these mutations or to intra-host recombination due to limits in the sensitivity of recombination analysis.”

Notwithstanding these limitations, our observations that some nonsynonymous mutations were observed across distinct haplotypes and that some of those haplotypes rose in frequency during the course of infection imply a beneficial effect of the mutations. We believe that this is the main biological interpretation to be made from these recurrent mutations.

Minor comments

Comment 1. P6/L127. "SARS-CoV-2 RNA levels were largely independent of diversity measures within subgroups (Extended Data Fig. 3d)." Upon initial review of Extended Data Fig. 3d, it appears that RNA levels (negatively correlated with Ct value) are positively correlated with genetic diversity, especially in the middle panel. Providing linear regression information would clarify this apparent discrepancy.

Author reply: Thank you for this question. As discussed above in response to the reviewer's *Major Comment 2*, we feel that the original Extended Data Fig. 3d (Supplementary Fig. 5 in the revised manuscript) had the potential to create confusion by presenting crude correlations between parameters that relate in complex ways. We have therefore excluded this analysis from the revised manuscript.

Comment 2. P12/L218. "We performed single-linkage phylogenetic clustering to identify clades of haplotypes in each person that were separated by at least 5 mutations and were thus likely to have originated from distinct founder". While the choice of 5 mutations is suitable for distinguishing different founder haplotypes, a comparison of donor-recipient haplotypes could offer a more accurate estimation of transmission founder size. Is there any available epidemiological information on transmission pairs among the patients in this study?

Author reply: We thank the reviewer for this question, and agree that a comparison of donor-recipient haplotypes would offer a more accurate estimation of transmission founder size than was achieved here. Unfortunately, there is no additional epidemiological information on transmission pairs in this study. We now acknowledge this limitation in the manuscript Discussion (lines 368-371), as follows:

“...defects associated with comorbid chronic infection could also promote acquisition of multiple SARS-CoV-2 variants through a relaxed transmission bottleneck, although verifying this speculation will require studies of SARS-CoV-2 transmission pairs.”

We would add that we performed founder analysis in part to detect multiple infection cases and thus allow proper partitioning of phylogenetic models when seeking to detect selection (i.e., to avoid treating branches linking distinct founder clades as intra-host evolution). We have further revised the section “Analysis of transmitted SARS-CoV-2 spike diversity” of the Results to highlight this for the reader on lines 218-221:

“To address this possibility and to limit the chance that downstream analysis would erroneously treat phylogenetic relationships between separate founder clades as intra-host evolution, we performed single-linkage phylogenetic clustering ...”

We do agree with the reviewer that a more accurate estimation of transmission founder size is an important goal for future studies, and emphasize that such studies are currently in development.

Comment 3. Fig. 1c y-axis label. "Number of mutation relative Hu-1". Add the preposition "TO" for clarity: "Number of mutations relative to Hu-1."

Author reply: We thank the reviewer for catching this and have revised the y-axis label in Fig. 1c.

Comment 4. Extended Data Fig. 4. The summary frequency of all possible haplotypes is expected to be approximately 1 for each sample. However, in several samples (e.g., 20-017, 20-015, 16-032), the summary frequency does not exceed 80%. Could this be attributed to numerous haplotypes with very low frequencies? Clarification on this aspect would be helpful.

Author reply: The reviewer is correct. Original Extended Data Fig. 4 (Supplementary Fig. 6 in the revised manuscript) showed only those haplotypes that were most abundant in the individual

in at least one timepoint throughout the infection. We agree that this was potentially confusing. For clarification on the interpretation of the figure, all haplotypes are now shown in stacked area plots, and haplotypes with lower frequencies are represented together in grey. We have also revised the figure legend accordingly.

Comment 5. Fig. 3 legend. "Fig.3. Analysis of intra-host spike mutations in PWH and PWOH." A missing space in "Fig.3". Correcting it to "Fig. 3" for consistency.

Author reply: We thank the reviewer for catching this. We added the space in “Fig. 3.”

Reviewer #2:

Summary: It has been known that SARS-CoV-2 can accumulate more mutations in people with HIV (PWH). In this study, Ko et al tried to understand the underline mechanisms by studying S sequences obtained by HT-SGS between PWH and PWOH populations. They show that haplotypes are more frequently detected in PWH than PWOH. However, there are some major concerns that need to be carefully addressed.

Major comments

Comment 1. The HT-SGS is not likely the real “SGS”. It is not clearly described in this study and the previous publication (I may have missed if it was) how the first round PCR was done. If it was carried as with bulk PCR (all templates together), there will be extensive PCR-derived artificial recombinants during amplification. This is well known in the field and the such artificial recombinants can account for up to 60% of the population. This will be especially worse for the amplicon as long as 4.3kb in this study. UMI will not resolve this problem. That is why the real SGS method has been developed by Coffin’s group also at NIH (PMID: 15635002) and widely used by others. At least the authors should use the Coffin SGS method to confirm their findings.

Author reply:

We thank the reviewer for calling attention to this central technical issue. Although we note that our original published manuscript describing HT-SGS does address technical recombinant sequences (see Fig. 1B and Table I, <https://doi.org/10.1371/journal.ppat.1009431>), we completely agree that the present manuscript should address the issue more fully, and should do so using the primers and sequencing target that apply to this study. Accordingly, we have now performed extensive additional control experiments to quantify technical recombinants in HT-SGS of spike, and have found that these account for <0.3% of all raw single-genome sequences in our data. This frequency is lower than that of the lowest reported haplotype in the present study. Thus, our new data should reassure the reader that our reported sequences are accurate. For additional details, please see our response below to the reviewer’s related **Comment 3**.

We do agree with the reviewer that single-copy amplification using limiting dilution in microtiter plates has long been a standard for intra-host virus genetic analysis, as described by the Coffin group and others. However, given our inclusion of control data demonstrating the high accuracy of our reported sequences, we believe that incorporating the Coffin group's method would not affect the findings or interpretations of our study. We would also note that limiting dilution PCR in plates is not practical for a study of >70,000 sequences.

Comment 2. The high numbers of haplotypes at the very early stage (3 or 4 DPSO) and persistent but similar levels of haplotypes throughout the studied period in PWH strongly indicate that these haplotypes were most likely the results of PCR-derived artificial recombinants during amplification. In addition, the haplotype profiles change dramatically just 1-2 days apart (Fig 4). Viral population should not change that fast.

Author reply:

As above in our reply to the reviewer's *Comment 1*, we appreciate the concern about technical recombinant sequences. For a fuller description of our new data that address this concern, please see our reply to the reviewer's *Comment 3*. These data are presented in a new supplementary section and make clear that PCR-derived and other types of technical recombinants do not account for our reported findings.

We also appreciate the reviewer's concern about the dramatic changes in haplotype profiles at timepoints spaced just 1-2 days apart, and agree that additional discussion of this issue is needed. One important limitation of the study that bears on this finding is the necessity to sample the intra-host virus population through mucosal swabs. A given swab sample likely represents only a portion of the virus genetic diversity present in the individual, and sampling identical positions within the mucosa at serial timepoints is not possible. Thus, it is likely that local differences in virus sequence populations within the mucosa of each person contributed to the observed dramatic changes in haplotype profiles at sequential timepoints. At the same time, while no swab sample can fully capture all SARS-CoV-2 sequences present in each person, the observed temporal fluctuations in haplotype profiles further highlight the high levels of diversity in participants with advanced HIV infection. We have now addressed these issues with the following revisions.

Results (lines 186-188): “In a longitudinal analysis of the most abundant spike haplotypes in each person, rapid fluctuations in detected frequencies observed in most PWH with CD4 counts <200 cells/ μ L (**Supplementary Fig. 6a**)....”

Discussion (lines 390-394): “Second, mucosal swab samples like those we analyzed here cannot fully capture the cumulative virus genetic diversity present throughout the body. Therefore, the numbers of spike sequence variants reported for each person in this study represent minimum estimates, and fluctuations in intra-host haplotype populations between samples may have been influenced by spatial variations in virus population composition.”

We also stress that any variations in haplotype profiles that may be associated with swab sampling variation were not sufficient to prevent identification of emerging variant sequences that support positive selection in people with advanced HIV infection.

Comment 3. They have obtained 70,968 sequences from 184 samples from 47 cases. However, it is not clear how many sequences are obtained for each time point from each case. How haplotypes are defined? Is one S gene with one mutation considered one haplotype? At what percentage a haplotype should be considered reliable?

Author reply:

We thank the reviewer for recognizing that the number of sequences obtained from each time point was not clearly provided. We have now added this information to Supplementary Data 2.

We also greatly appreciate the reviewer’s request that we more clearly establish the validity of haplotypes reported in the study. As described above, we have now performed extensive additional control experiments to address this. Results of these experiments are presented in a new supplementary section, and include the following:

1. *False variant analysis using clonal RNA.* Clonal RNA samples generated by *in vitro* transcription of plasmid constructs were analyzed by HT-SGS, which includes our custom

bioinformatic pipeline. These experiments did not detect any mutations when considering resulting datasets either in their entirety or in 1000 random subsamples of final SGS. Thus, these experiments established an estimated false discovery rate (FDR) of <1 false variant per 1000 samples processed by our method (Supplementary Fig. 3), reflecting our conservative thresholds for reporting sequences detected in HT-SGS. Simultaneously, our approach was able to resolve titrated mutation mixtures precisely to frequencies of ~0.2% (Supplementary Fig. 3).

2. *Technical recombinant analysis using mixtures of clonal RNAs.*

- a. *Recombinant SGS.* Using a pair of synthetic spike constructs in which one construct had two anchor mutations to facilitate detection of recombination (i.e., mutations near the 3' and 5' end of the target amplicon, see Supplementary Fig. 4), we again subjected *in vitro* transcribed RNA to HT-SGS. Out of 831 SGS recovered across 8 samples, only 1 recombinant was detected (corresponding to an estimated recombination rate at the SGS-level of 0.24%, see Supplementary Fig. 4). This resembles findings from our previous publication that did not detect any recombinant SGS across more than 700 sequences from a longer amplicon. We also note that, because our pipeline only includes haplotypes detected in >1 SGS, the 1 recombinant sequence detected was not called as a haplotype.
- b. *Recombinant sequence reads within non-recombinant SGS "bins."* Although PCR-mediated recombinant reads that account for only a minority of reads within a given SGS/UMI bin do not cause errors in our final reported data, we recognize that presenting an analysis of such reads may nonetheless be of interest to the reader. As now shown Supplementary Fig. 4, PCR-derived recombinant reads rarely accounted for >1% of reads within each bin.
- c. *Recombinant sequence reads within non-recombinant SGS bins in experiments with whitelisted UMIs.* Out of an abundance of caution, we validated our results in parallel using a custom RT primer pool with 354 defined UMI sequences (i.e., whitelisted UMIs). The use of whitelisted UMIs ensures unambiguous read binning when analyzing UMI sequences in raw sequencing data. The results of these experiments matched the results from the 8N UMI approach used in the rest of the

study (see Supplementary Fig. 4). This further supports that recombination-mediated artifacts do not corrupt our haplotype sequences or the biological findings reported in our manuscript.

In view of these data, we provide the following definition of a valid haplotype: haplotypes are defined as unique combinations of **called variants** within single-genome sequences that are seen **in multiple SGS**. Thus, a spike sequence with one mutation would meet the criteria for haplotype calling if 1) the mutation comprising it was seen above the background level of expected SGS-level errors and 2) two identical SGS with the mutation were detected. We have added earlier in the manuscript a mention that haplotypes considered for analysis are sequences called from an analysis of SGS, not the SGS themselves, see section “HT-SGS vs. standard whole-genome sequencing”: “By considering the unique combinations of called mutations (i.e., haplotypes) that are supported by multiple single-genome sequences, HT-SGS demonstrates ...”

As stated above, these data and arguments have been added to a new supplementary section of the manuscript, and are summarized in the revised section ‘HT-SGS vs. standard whole-genome sequencing (lines 100-102 and 105-108), as follows:

“Control experiments using mixtures of synthetic RNAs supported the validity of minor variant spike sequences detected at this level in HT-SGS data (**Supplementary Fig. 3**)... Additional controls using mixtures of synthetic RNAs verified that haplotypes detected by HT-SGS accurately represented sequences present in sample material, without significant evidence of technical recombination artifacts (**Supplementary Fig. 4**).”

We hope that these additions to the manuscript will clarify that the haplotypes and associated measurements of genetic diversity reported in the study are biological findings and are not the result of PCR-mediated recombination events.

Comment 4. Do the phylogenetic trees include all the sequences from each case in Fig. 4? If so, are there identical sequences in each case? The sequences from different time points should be clearly indicated in the tree. Do viruses, not haplotypes, evolve over time? What are the diversity

levels from different time points in each case? Does it change over time? These are the key information that should be presented.

Author reply: We thank the reviewer for this question. The phylogenetic trees show all haplotypes from each case, with each leaf on each tree defined by one haplotype. The frequency of each haplotype at each timepoint is indicated by the dot plot to the right of the tree. To better illustrate the population changes over time, we have added Muller diagrams to the Supplementary Fig. 12 which include measurements of genetic diversity (normalized Shannon entropy) and autologous spike binding titers with each plot. We believe that these visualizations display the key information necessary to capture the evolution occurring in our data and thank the reviewer for highlighting the need to integrate these data visually.

Comment 5. The all groups are not comparable. The time for the PWOH, PWH (CD4 \geq 200) and PWH (CD4 unknown) were very short, while PWH (CD4 <200) were followed much longer time. It is very likely that the time is the determining factor. Is the sequence for each case in Fig. 3 the accumulation all detectable mutations? If so, this may be misleading since some very rare ones will carry the same weight as those predominant ones.

Author reply: We appreciate the reviewer's comment, which raises two separate issues.

1. The initial submitted Fig. 3 showed all detectable mutations, without distinguishing very rare ones from predominant ones. We agree that this had the potential to cause confusion. Therefore, our revised Fig. 3a distinguishes mutations by maximum measured frequency (i.e., mutations whose maximum frequencies were \geq 20%, \geq 5%, or <5% of all SGS in a given timepoint). We also moved the section of Fig. 3a showing mutations in PWOH to the Supplementary Information in the revised manuscript so that mutations in PWH could be displayed more clearly in the main text figure (see new Supplementary Fig. 8a).
2. Sequence analysis spanned longer intervals PWH with CD4 <200 than in the other participants, complicating comparisons among groups. We understand this concern. In our revised manuscript, we have emphasized that all participants were followed with

prospective swab samples until virus clearance, as follows (lines 248-253 in ‘Spike mutations in PWH and PWOH’ section):

“Although several PWH with higher CD4 counts and PWOH also showed nonsynonymous intra-host mutations (**Fig. 3a, b; Supplementary Fig. 8a**), the number of nonsynonymous intra-host mutations per person was significantly higher in PWH with CD4 counts <200 cells/ μ L than in the other subgroups (**Fig. 3d**), likely a reflection of higher levels of virus replication and longer durations of infection in this subgroup.”

Thus, differences in sampling duration between groups reflect real differences in shedding duration, and the SARS-CoV-2 genetic findings in PWH with CD4 <200 therefore reflect the distinctive biology in these individuals.

It is furthermore important to address the reviewer’s comment that time is the determining factor in SARS-CoV-2 spike genetic diversity differences between PWH with CD4 <200 and other groups. While we agree that longer infection duration must certainly contribute to higher diversity in PWH with CD4 <200, we believe it is also valid to stress that diversity was elevated in PWH with CD4 <200 even at the first sampling timepoint. Thus, longer duration of infection may not be the only determining factor of SARS-CoV-2 spike genetic diversity differences among groups in our study. This issue has important biological implications, and is addressed in more detail in response to the reviewer’s *Comment 6*.

Comment 6. It is surprising to see high level variability only 3 or 4 DPSO in PWH. What caused this? This cannot be explained by poor immunity in PWH. SARS-CoV-2 does not change much in cell culture.

Author reply: We agree that our finding of high SARS-CoV-2 spike diversity at only 3 or 4 DPSO in PWH with CD4 <200 is surprising, and feel it is an important component of the biological narrative that deserves thorough exploration. In theory, causes could include faster generation of diversity due to a higher replication rate and/or higher transmitted diversity due to a relaxed transmission bottleneck in the setting of attenuated host defenses in PWH with CD4 <200. We

now discuss these possibilities in greater detail in the revised manuscript, as follows (lines 364-371 in ‘Discussion’ section).

“Of particular interest, elevated spike gene diversity detected in people with advanced HIV infection even before the expected onset of adaptive immunity could indicate innate immune defects in these individuals. Such defects may include impairment of type I interferon responses, which could allow SARS-CoV-2 to diversify rapidly through a reduced barrier to its replication. Impaired type I interferon responses or other defects associated with comorbid chronic infection could also promote acquisition of multiple SARS-CoV-2 variants through a relaxed transmission bottleneck, although verifying this speculation will require studies of SARS-CoV-2 transmission pairs.”

Regarding the reviewer’s statement that SARS-CoV-2 does not change much in cell culture, we feel it is appropriate to note extensive reports of furin cleavage site deletions in early work on viruses passaged in cell culture [1-3]. These reports include our own publication, which demonstrated 17 minor variant haplotypes in a 4th-passage culture of the WA-1 clinical isolate (see Fig. 2 in the reference 8). We believe that, in retrospect, these findings are understandable, given that growing a virus adapted to the human respiratory tract in culture almost certainly represents a significant change in the fitness landscape.

Reference for the Comment 6

1. M. M. Lamers et al., “Human airway cells prevent SARS-CoV-2 multibasic cleavage site cell culture adaptation,” *eLife* **10**, e66815 (2021).
2. D.-Y. Chen et al., “Cell culture systems for isolation of SARS-CoV-2 clinical isolates and generation of recombinant virus,” *iScience* **26**, 106634 (2023).
3. S. H. Ko et al., “High-throughput, single-copy sequencing reveals SARS-CoV-2 spike variants coincident with mounting humoral immunity during acute COVID-19,” *PLOS Pathogens* **17**, e1009431 (2021).

Comment 7. The abstract is misleading. The study did not really show “rapid emergency and evolution of SARS-CoV-2”

Author reply: We thank reviewer for this comment. Our findings in this study demonstrate that, in PWH with CD4 counts <200 but not in other participant groups, many SARS-CoV-2 variants were detected within 1 or 2 weeks (Fig. 1a and Fig. 4), and genetic diversity immediately after COVID-19 symptom onset was high (Fig. 2a). Founder sequences rapidly declined over time (Fig. 2d) and composition of spike sequence population fluctuated considerably over time (original Extended Data Fig. 4a (Supplementary Fig. 6a in the revised manuscript)) in PWH with CD4 <200. We understand that part of the reviewer’s initial concern may have been related to the technical validity of our reporting findings. Given that we have now provided data from control experiments addressing this issue, we hope that the reviewer will agree with the statements expressed in our newly revised title and abstract, as supported by our revised main text, main figures, and extended data.

Comment 8. Other regions should also be studied. At least one region from a few cases.

Author reply: We appreciate reviewer’s comment and acknowledge that incorporating other genomic sequences beyond the spike gene could be informative. Nonetheless, called haplotypes of the full-length spike gene that differ by at least 1 base must represent distinct whole-genome haplotypes. In other words, if two genomes have different spike sequences, they must represent different intra-host variants. Our study uses spike sequences to find marked differences in genetic diversity between PWH with CD4 <200 and other groups of participants. These findings and their potential mechanisms discussed in the manuscript represent the central message of the study. In the revised manuscript, we now acknowledge the limitation of spike-only sequencing while explaining that the choice to sequence spike was related to this gene’s relatively high diversity (see lines 384-389 in Discussion section).

“Our findings in this study have several limitations. First, although our HT-SGS technology combines high accuracy with relatively deep sampling, enzyme processivity limitations in reverse-transcription (RT) and PCR prevent the efficient preparation of single-copy libraries

for amplicons longer than several kilobases. While we chose to sequence spike due to a relatively high genetic diversity combined with acceptable SGS yield, we acknowledge that future technical advances may enable single-copy sequencing of longer targets including informative regions outside spike.”

Comment 9. The R values are too low to be meaningful in Fig. 2.

Author reply: We thank the reviewer for this comment. The goal of the statistical analysis in Fig. 2b is to determine whether there is a significant correlation between initial SARS-CoV-2 genetic diversity (3 ~ 4 DPSO) and shedding duration. Such a relationship could be either linear or non-linear. Therefore, we have now performed this analysis using Spearman’s rank correlation test, rather than linear regression, with new R values presented in the revised Fig. 2b. We hope that the statistical significance of the resulting findings will be sufficient to support our claim of a relationship between initial SARS-CoV-2 diversity and shedding duration.

Comment 10. At what levels, two founders were defined?

Author reply: The criteria for identification of multiple founder sequences was a single-linkage minimum distance of 5 mutations between phylogenetic clades (see lines 220-222 in ‘Analysis of transmitted SARS-CoV-2 spike diversity’ section, and section “Phylogenetic inferences” of Methods). These criteria enabled us to identify two cases of evident multiple infection (S006-001 and S074-001). Finding these cases among the group of PWH with CD4 <200 is potentially consistent with the idea of a looser transmission bottleneck in these individuals, although this possibility will require further investigation, as acknowledged in the revised Discussion section (lines 368-371):

“...other defects associated with comorbid chronic infection could also promote acquisition of multiple SARS-CoV-2 variants through a relaxed transmission bottleneck, although verifying this speculation will require studies of SARS-CoV-2 transmission pairs.”

Reviewer #3:

Summary: This study focuses on the within-host evolutionary kinetics of SARS-CoV-2 in people with and without HIV (controlled and poorly controlled). The study finds that in people with poorly controlled HIV, SARS-CoV-2 evolution is marked by rapid diversification by the time of symptom onset and higher diversity of the within-host virus population as compared to people without HIV. The manuscript is very well written and was a joy to read. The approach, methods, and conclusions are all clear and well justified.

Author reply: We thank the reviewer for these positive comments.

Major comments

Comment 1. My sole critique relates to the first paragraph of the introduction - Line 55 – “However, convergent evolution of the same mutations in unrelated persistent cases implies extensive early intra-host SARS-CoV-2 sequence diversification that has not been directly observed. Many persistent infections described in previous studies have been characterized retrospectively, with limited analysis during the acute phase.” – This feels like it is being put forth as the unique selling point for this story but I don’t follow the logic here. This could also be the case if there are limited high fitness evolutionary paths. And capturing the acute phase is fine, but the real strength of this study is the number of study participants allowing for more robust conclusions and a more authoritative presentation of results. I suggest highlighting this point.

Author reply: We greatly appreciate the reviewer’s comment. We have now removed the above quoted statements from the Introduction, and have revised the first paragraph of the Introduction to highlight the strength of our study as follows (see lines 53-55 in Introduction section).

“...many previous studies retrospectively characterizing persistent infections have been limited to case reports or small case series, and have not closely examined the early stage of infection.”

Reviewers' Comments:

Reviewer #1:

Remarks to the Author:

Hi, I'm the previous Reviewer #1. The authors have addressed my comments comprehensively, providing detailed explanations and making necessary revisions to the manuscript. Their responses are well-argued and demonstrate a thorough understanding of the concerns raised. The additional analyses and visualizations they included enhance the clarity and robustness of their findings.

I have only one minor comment on the observed adaptive selection and higher diversity in PWH with advanced infection. According to population genetics principles, selection (selective sweep or purifying selection) typically reduces diversity in the population. Intra-host positive selection is identified almost exclusively among PWH with CD4 counts <200 cells/ μ L. Why is diversity higher among these participants compared to other groups? The authors attribute this to the complexity of the biology, such as replication limitation in people with healthy immune systems. However, this potential force should be represented by the selection signal (dN/dS) within the host.

A more plausible explanation is the antagonistic pleiotropy of mutations intra- and inter-host (PMC10521905, PMC11021471). Specifically, some mutations may have no fitness gain within the host but could provide fitness advantages during transmission between hosts. Thus, neutral or slightly deleterious mutations within the host may not be easily swept away, maintaining higher diversity. The authors should consider discussing these cases in the Discussion section.

Overall, the revisions have significantly improved the manuscript, making it suitable for publication.

Reviewer #2:

Remarks to the Author:

The authors have addressed my comments. I only have a couple of comments which the authors may want to elaborate further.

I am still amazed that the genetic diversity is so much higher in PWH (CD4 <200) than in PWH (CD4 ≤ 200) and PWOH during the first few days of infection. This cannot be due to the adaptive immunity. If this is likely caused by the innate immunity (e.g., type I interferon responses as the author speculated), is there any evidence for innate immunity to so rapidly impact the genetic diversity on SARS-CoV-2 or any other viruses?

Both synonymous and nonsynonymous mutations are much higher for PWH (CD4 <200) than PWH (CD4 ≥ 200) or PWOH (Figure 3C and 3D). This indicates that the high level of diversity in PWH (CD4 <200) is not likely due to selection pressure from the adaptive or innate immunity. Can authors discuss this scenario?

Reviewer 1

Hi, I'm the previous Reviewer #1. The authors have addressed my comments comprehensively, providing detailed explanations and making necessary revisions to the manuscript. Their responses are well-argued and demonstrate a thorough understanding of the concerns raised. The additional analyses and visualizations they included enhance the clarity and robustness of their findings.

I have only one minor comment on the observed adaptive selection and higher diversity in PWH with advanced infection. According to population genetics principles, selection (selective sweep or purifying selection) typically reduces diversity in the population. Intra-host positive selection is identified almost exclusively among PWH with CD4 counts <200 cells/ μ L. Why is diversity higher among these participants compared to other groups? The authors attribute this to the complexity of the biology, such as replication limitation in people with healthy immune systems. However, this potential force should be represented by the selection signal (dN/dS) within the host.

A more plausible explanation is the antagonistic pleiotropy of mutations intra- and inter-host (PMC10521905, PMC11021471). Specifically, some mutations may have no fitness gain within the host but could provide fitness advantages during transmission between hosts. Thus, neutral or slightly deleterious mutations within the host may not be easily swept away, maintaining higher diversity. The authors should consider discussing these cases in the Discussion section.

Author reply

We thank the reviewer for these thoughtful and interesting comments. If understood correctly, the comments suggest that elevated SARS-CoV-2 diversity in people with advanced HIV infection is determined largely at transmission, and that clearance of high transmitted diversity in these individuals may not be possible due to antagonistic pleiotropy of mutations. Though we prefer the view that elevated SARS-CoV-2 diversity in people with advanced HIV infection arises largely by mutations within the individual (rather than variants already present at transmission), we acknowledge that the reviewer's suggestion cannot be ruled out based on our data because it is not possible to know with certainty which sequences detected at the first sample timepoint were present at transmission.

We now acknowledge this issue in the revised Discussion section, as follows:

"Third, because our study relied on natural infections, we are unable to determine the timing of SARS-CoV-2 transmission to our study participants. We cannot unambiguously determine which SARS-CoV-2 variants detected in each person were present at transmission...."

We believe that future studies, perhaps using household transmission clusters or experimental infections in model systems, will lead to a better understanding of this important issue.

Reviewer 2

The authors have addressed my comments. I only have a couple of comments which the authors may want to elaborate further.

I am still amazed that the genetic diversity is so much higher in PWH (CD4<200) than in PWH (CD4≤200) and PWOH during the first few days of infection. This cannot be due to the adaptive immunity. If this is likely caused by the innate immunity (e.g., type I interferon responses as the author speculated), is there any evidence for innate immunity to so rapidly impact the genetic diversity on SARS-CoV-2 or any other viruses?

Both synonymous and nonsynonymous mutations are much higher for PWH (CD4<200) than PWH (CD4≥200) or PWOH (Figure 3C and 3D). This indicates that the high level of diversity in PWH (CD4<200) is not likely due to selection pressure from the adaptive or innate immunity. Can authors discuss this scenario?

Author reply

We appreciate (and share) the reviewer's amazement at the high levels of diversity we detected in PWH with CD4<200 during the first few days of infection. Although we are not aware of any studies that have correlated early SARS-CoV-2 diversity with innate immune status or function, we did note that innate immune stimulation with exogenous interferon-alpha was found to reduce the number of transmitted virus variants in a non-human primate challenge study with simian immunodeficiency virus (SIV) (PMC4418221).

Furthermore, we agree with the reviewer that the pattern of mutations in these individuals – including both synonymous and nonsynonymous mutations – does not support selection as the only mechanism driving elevated diversity. Indeed, we believe that an early failure to control SARS-CoV-2 replication (and/or a widened transmission bottleneck) is a critical piece of the story. We now address this in greater detail in the final paragraph of the Discussion section, as follows.

“We propose that these differences [in diversity, between PWH with advanced HIV and others] stem from multifactorial impairments in antiviral defenses during advanced HIV infection, which 1) enable elevated early SARS-CoV-2 replication (and, thus, SARS-CoV-2 genetic diversification) and 2) present SARS-CoV-2 with a relatively low barrier to epitope-specific mutational escape. By contrast, the virus may have less opportunity to diversify in people with well-controlled or no HIV infection, while stronger, more rapid, and potentially broader immune responses in these individuals create a relatively large barrier to escape.”